# OpsEval: A Comprehensive Benchmark Suite for Evaluating Large Language Models' Capability in IT Operations Domain

## Abstract

The past decades have witnessed the rapid development of Information Technology (IT) systems, such as cloud computing, 5G networks, and financial information systems. Ensuring the stability of these IT systems has become an important issue. Large language models (LLMs) that have exhibited remarkable capabilities in NLP-related tasks are showing great potential in AIOps, such as root cause analysis of failures, generation of operations and maintenance scripts, and summarizing of alert information. Unlike knowledge in general corpora, knowledge of Ops varies with the different IT systems, encompassing various private sub-domain knowledge, sensitive to prompt engineering due to various sub-domains, and containing numerous terminologies. Existing NLP-related benchmarks (e.g., C-Eval, MMLU) can not guide the selection of suitable LLMs for Ops (OpsLLM), and current metrics (e.g., BLEU, ROUGE) can not adequately reflect the question-answering (QA) effectiveness in the Ops domain. We propose a comprehensive benchmark suite, **OpsEval**, including an Ops-oriented evaluation dataset, an Ops evaluation benchmark, and a specially designed Ops QA evaluation method. Our dataset contains 7,334 multiple-choice questions and 1,736 QA questions. We have carefully selected and released 20% of the dataset written by domain experts in various sub-domains to assist current researchers in preliminary evaluations of OpsLLMs. We test over 24 latest LLMs under various settings such as self-consistency, chain-of-thought, and in-context learning, revealing findings when applying LLMs to Ops. We also propose an evaluation method for QA in Ops, which has a coefficient of 0.9185 with human experts and is improved by 0.4471 and 1.366 compared to BLEU and ROUGE, respectively. Over the past one year, our dataset and leaderboard have been continuously updated.

## 1 Introduction

The IT Operations (Ops) plays a crucial role in maintaining the efficient and stable operation of information systems such as cloud computing, 5G networks[1] and financial information systems. As the Internet continues to expand rapidly, the scale and complexity of systems are escalating, leading to the emergence of artificial intelligence-assisted operations as a novel trend. Termed "AIOps" by Gartner (Lerner, 2017), this technique utilizes artificial intelligence to address tasks such as anomaly detection, fault analysis, and performance optimization. In recent years, large language models (LLMs) have witnessed significant advancements. The latest models, such as GPT-4o (OpenAI, 2024), GPT-4V (OpenAI, 2023b), Meta-Llama-3 (AI@Meta, 2024), and GLM-4 (Zeng et al., 2022), have demonstrated exceptional generalization and task-planning capabilities. As a result, these models have provided numerous opportunities to enhance downstream domain-specific applications. With its advanced text generation ability, LLM is well suited for Ops on tasks like question answering, information summarizing, and report analysis. Hereinafter, we refer to the LLM used for Ops as **OpsLLM**, regardless of whether they have been optimized specifically for Ops.

---

[1]Strictly speaking, 5G belongs to the field of communications technology (CT), but given its broad association with the information technology (IT) sector, for the sake of generality, we refer to it as IT operations, abbreviated as Ops, throughout the remainder of this paper.

While there are benchmarks for assessing general-purpose NLP-related capabilities, no benchmark exists to evaluate the effectiveness of LLMs or OpsLLMs in Ops tasks. There is an urgent need for an Ops benchmark that informs us about the performance of current LLMs on Ops tasks. On the other hand, a good benchmark can significantly aid the optimization process of OpsLLMs tailored for the Ops domain. Nevertheless, due to the specialty of the Ops tasks, constructing an Ops benchmark presents the following challenges:

**1) Sensitive data.** The Ops data is primarily sensitive and proprietary to companies, with very few publicly available data, making it difficult for any company to independently provide sufficient evaluation data to ensure confidence in the test results. **2) Sub-domains.** The Ops field spans many sub-domains, like 5G communications, cloud computing, and bank transactions, each requiring a mix of capabilities, or "tasks," such as network configuration or terminology explanation. The sheer number of sub-domains and tasks, combined with the absence of a systematic taxonomy, makes classifying questions challenging. **3) Prompt sensitivity.** Due to the relatively proprietary nature of the Ops, existing LLMs have not undergone specialized supervised fine-tuning (SFT) for instruct following within the Ops field, the evaluation results are more sensitive to prompt engineering. Designing appropriate prompts for robust and accurate evaluation is challenging. **4) QA metric.** Existing metrics like BLEU focus on linguistic similarity between model output and reference answers, which often fails to capture true performance in Ops tasks. In Ops, it's essential to assess whether the model's answers address key points in the reference and are supported by sufficient evidence, reflecting the precise meanings of domain-specific terms.

To address these issues, we propose **OpsEval**, a comprehensive benchmark suite for evaluating LLMs' capability in the IT operations domain. First, to tackle the challenge of benchmark data mostly being private, we initiated a community around AIOps, which has attracted dozens of companies to participate. We have selected 9 representative sub-domains from the community, allowing continuous data contributions from community members. We then aggregate data under the same sub-domain to ensure robustness in evaluation. Additionally, we generated multi-choice (MC) and question-answering (QA) questions as supplements based on publicly available network management books. To address the challenge of classifying the numerous sub-domains and tasks in the Ops field, we employ model-based pre-clustering and manual review to annotate eight tasks and three abilities. Considering the prompt sensitivity of benchmark results, we systematically test model performance under self-consistency (SC), chain-of-thought (CoT), and few-shot in-context learning (ICL). Lastly, to address the inaccuracy of existing metrics in Ops QA evaluation, we design FAE-Score, which evaluates model responses based on fluency, accuracy, and evidence, with each criterion having its own dedicated assessment method.

The contributions of our paper are as follows: **1)** We introduce **OpsEval**, the first bilingual multi-task dataset in the Ops domain, covering 8 tasks and 3 abilities with 9,070 questions. To assist researchers in preliminary evaluating their OpsLLMs, we have carefully selected and released 20% of QAs from our benchmark licensed under CC-BY-NC-4.0, with the remaining 80% of undisclosed data preventing unfair evaluations due to data leakage (Wei & et.al., 2023) **2)** Based on the dataset, we introduce the OpsEval evaluation benchmark, conducting independent and robust evaluations with various prompting techniques and a specifically designed evaluation metric, FAE-Score. Compared to the commonly employed BLEU and ROUGE metrics, FAE-Score exhibits a more pronounced congruence with the evaluations of human experts. Specifically, FAE-Score attains a correlation coefficient 0.9175 with expert assessments, surpassing the coefficients of 0.6705 for BLEU and -0.3957 for ROUGE. **3)** Based on the results of OpsEval evaluation, we provide key observations and practical lessons to help domain practitioners make decisions such as whether existing models are sufficiently applicable within a specific sub-domain, the necessity for fine-tuning and whether model quantization compromises the effectiveness.

## 2 RELATED WORKS

As LLMs evolve rapidly, their complex and varied capabilities are increasingly recognized. As a result, there is a growing trend towards evaluation benchmarks tailored specifically for LLMs. These can be divided into two categories: general ability benchmarks and domain-specific benchmarks.

**General ability benchmarks** assess the general abilities of LLMs across various tasks. These tasks evaluate LLMs' capacity for logical reasoning, general knowledge, common sense, and other simi-

Table 1: A comparison of OpsEval with other popular datasets/benchmarks.

| | MMLU | HELM | BIG-bench | SEAL | CEval | FLUE | MultiMedQA | CMB | NetOps | OWL | OpsEval |
|---|---|---|---|---|---|---|---|---|---|---|---|
| Ops Domain Dataset | ✗ | ✗ | ✗ | ✓ | ✓ | ✗ | ✗ | ✗ | ✓ | ✓ | ✓ |
| Open-sourced Benchmark | ✓ | ✓ | ✓ | ✗ | ✓ | ✓ | ✓ | ✓ | ✗ | ✗ | ✓ |
| Up-to-date Leaderboard | ✓ | ✓ | ✓ | ✓ | ✓ | ✗ | ✗ | ✗ | ✗ | ✗ | ✓ |

Figure 1: The framework of OpsEval

lar abilities rather than being confined to a particular domain. MMLU (Hendrycks et al., 2021) is a benchmark designed to measure knowledge acquired during pretraining by evaluating models exclusively in zero-shot and few-shot settings, covering 57 subjects across STEM. HELM (Liang et al., 2022) employs seven distinct metrics in 42 unique scenarios, offering a comprehensive evaluation of LLMs' capabilities across multiple dimensions. BIG-bench (Srivastava et al., 2022) comprises 204 tasks spanning a wide array of topics, with a particular focus on tasks deemed beyond the reach of current LLMs. SEAL (AI, 2024b) features private, expert evaluations of leading frontiers models. C-Eval (Huang et al., 2023) is a comprehensive Chinese evaluation suite designed to assess Chinese LLMs' advanced knowledge and reasoning abilities rigorously.

**Domain-specific benchmarks** evaluate the abilities of LLMs to handle tasks in specific fields. These benchmarks require LLMs to possess specialized knowledge in a specific domain and to respond in a manner consistent with the cognitive patterns of that field. Despite the rapid progression of LLMs in specialized domains, the evaluation metrics for these specific areas have received less attention. FLUE (Shah et al., 2022) is an open-source comprehensive suite of benchmarks, including new benchmarks across 5 NLP tasks in financial domain. MultiMedQA (Singhal et al., 2022) is an extensive medical question-answering dataset, with questions derived from professional medical exams, research, and consultation records. CMB (Wang et al., 2023a) includes multi-choice questions (CMB-Exam) and complex clinical questions based on real case studies (CMB-Clin). NetOps (Miao et al., 2023) focuses on evaluations in the network field, which is relevant to the field of Ops. NetOps includes multi-choice questions in both English and Chinese and a few question-answering questions. However, they only focus on wired network operations and while the dataset is released, they lack a benchmark that continuously updates the leaderboard. OWL (Guo et al., 2024) introduces Owl-Instruct and Owl-Bench datasets for IT operations, along with methods like HMCE for handling input length and a mixture-of-adapter for efficient tuning. However, it lacks a real-time updated leaderboard and does not provide a well-designed evaluation for IT operations QA tasks.

## 3 OPSEVAL BENCHMARK

Figure 1 shows the overall framework of OpsEval from construction to evaluation. We collected data from multiple sources and then preprocessed it to enhance its quality. Finally, we evaluated LLMs on the dataset using various prompt engineering techniques.

### 3.1 DATA COLLECTION

Our benchmark questions have been collected from various sources; we summarize them into four categories: company materials, certification exams and Ops textbooks. Each source is highly esteemed globally and reviewed by our Ops collaborators.

**Company Materials.** include production environment materials like Ops tickets and error logs , as well as internal documents and tests for Ops staff training. We have established cooperative relationships with 11 companies, covering various sectors like telecommunications, finance, and Ops service/tool providers, and received expert collaboration and Ops materials from them. The Appendix A.1 provides information about the companies and experts.

Table 2: Overview of the question distribution in OpsEval by sub-domains, tasks and abilities.

(a) The number of questions in OpsEval, grouped by their sub-domains.

| Sub-domain | Source | Type | Questions |
|---|---|---|---|
| Wired Network | Operation Textbooks | MC | 3901 |
| 5G Communication | Certification Exams | MC | 2615 |
| | | QA | 1162 |
| Oracle Database | Company Materials | MC | 497 |
| Log Analysis | Company Materials | QA | 420 |
| DevOps | Company Materials | QA | 154 |
| Private Cloud | Company Materials | QA | 150 |
| Securities Info. | Company Materials | MC | 91 |
| Hybrid Cloud | Company Materials | MC | 40 |
| Financial IT | Company Materials | MC | 40 |
| Total | | | 9,070 |

(b) The distribution of different tasks and abilities of questions in OpsEval.

| | Category | Percentage (%) |
|---|---|---|
| **Task** | Automation Scripts | 3.3 |
| | Monitoring and Alerting | 5.2 |
| | Performance Optimization | 5.3 |
| | Software Deployment | 7.9 |
| | Fault Analysis and Diagnostics | 13.7 |
| | Network Configuration | 29.0 |
| | General Ops Knowledge | 20.2 |
| | Miscellaneous | 15.5 |
| **Ability** | Knowledge Recall | 49.8 |
| | Analytical Thinking | 39.9 |
| | Practical Application | 10.2 |

**Certification Exams.** include knowledge assessments necessary for becoming an Ops staff and are naturally in the form of multiple-choice and question-answering questions. We obtained the relevant study guidebooks for these certification exams from public book websites and extracted sample questions from them as one of the sources for Ops questions.

**Operations Textbooks**. We first constructed a seeding keyword list for the Ops field and searched for related books. The textbooks contain relatively complete knowledge content, which can provide experts with materials for question creation, and some books themselves also include a certain number of exercises at the end of the chapters.

## 3.2 PREPROCESSING

We systematically carried out the preprocessing of our original data in the following stages:

**Deduplication:** Any repeated or highly similar questions are identified and removed to avoid redundancy in the test set. We calculate the cosine similarity of the question stems by bge-large-zh-v1.5 (Xiao et al., 2023) to detect duplicate questions and identify pairs of questions with a similarity above a certain threshold (th=0.7).

**Dependance Filtering:** We have filtered out questions that rely on external images or document content to ensure the completeness of the question content itself. The filtering process was done by two parallel lists of empirical keywords in the question stems and the responses of GPT-3.5-turbo. The keyword list can be found in the Appendix A.2.

**Question Categorization:** We devise a categorization that captures many tasks that professionals confront in practical applications. The categorization process consists of two steps: automated screening and manual review. We first use GPT-4 for topic modeling to gain rough insights about the dataset and determine the relevance of each question to Ops, which resulted in more than 20 tasks but had an imbalanced distribution. We then involved dozens of experts during the manual review process to categorize the questions into eight tasks and three abilities. The distribution of the questions across these eight tasks and three ability levels is shown in Table 2b, and the details of each task and ability can be found in Appendix A.4.

**Manual Review:** In the manual review step, we asked Ops experts from the industry to inspect the results of the previous three automated steps, including confirming duplicate and invalid questions and examining the classification results of GPT-4. In our work, an expert is defined as an individual with ten or more years of professional experience in their field, whether as an employee or a researcher. Experts were also asked to drop the questions unrelated to the Ops field. We split the dataset by n-folds and ensure each fold has at least two experts to review. As listed in Table 2a, this quality enhancement process resulted in a refined test set of approximately 7,000 multi-choice and 2,000 question-answering questions.

## 3.3 EVALUATION SETTINGS

**Multi-choice questions** offer a structured approach with definitive answers. These questions are straightforward and provide a clear metric for assessment. We use **accuracy** as the metric. A

choice-extracting function based on regular expressions is used to extract the predicted answer of LLMs. Then, we calculate the accuracy based on the extracted answer and the ground-truth labels.

**Question-answering questions.** We evaluate question-answering tasks using a metric designed specifically for OpsEval, called **FAE-Score**, which is explained in detail in the subsequent section. Additionally, we perform expert evaluations and calculate BLEU (Papineni et al., 2002), ROUGE (Lin, 2004) and RAGAS (Es et al., 2024) scores for comparison purposes, as reference to validate the accuracy of FAE-Score.

We use the same three criteria to evaluate the responses of various models for both FAE-Score and Expert Evaluation:

- **Fluency**. Assessment of the linguistic fluency in the model's output and compliance with the question-answering question's answering requirements.

- **Accuracy**. Evaluation of the precision and correctness of the model's output, including whether it adequately covers key points of the ground-truth answer.

- **Evidence**. Examine whether the model's output contains sufficient argumentation and evidential support to ensure the credibility and reliability of the answer.

In Expert Evaluation, we asked experts to score it between 0 and 3 for each criterion. During the scoring, the raw question, the detailed answer and its key points, and the output of an anonymous model are given at each iteration.

**Prompting Techniques.** We use various settings to evaluate LLMs on OpsEval to get a comprehensive overview of their performance. We evaluate LLMs in zero and few-shot (3-shot) settings. For each setting, we evaluate LLMs in four sub-settings of prompt engineering, that is, naive answers (Naive), self-consistency (SC) (Wang et al., 2023b), chain-of-thought (CoT) (Wei et al., 2023), self-consistency with chain-of-thought (CoT+SC). We set the number of queries in SC to **5**.

**Models.** We evaluate popular LLMs covering different weights from different organizations. The model selection was guided by specific criteria: We aimed to include the latest and most advanced large language models, with a particular focus on those capable of handling Chinese input. The detailed information of all 24 LLMs can be found in Table 6 in Appendix C.1.

### 3.4 FAE-SCORE

Figure 3 shows the basic pipeline of our designed QA metric, FAE-Score. Here, we elaborate each evaluation methodology of each criterion.

**Fluency.** In Ops settings, the fluency of a model's output is crucial because the results are intended for human consumption by technical personnel. Unlike other generic benchmarks, the tasks in the Ops domain often require clear and unambiguous communication, as the model's outputs may guide decision-making in real-world scenarios. Therefore, ensuring high fluency in responses is not just a matter of language quality but a critical factor for task completion and user comprehension. To evaluate fluency in model outputs, we adapted the scoring rubrics methodology mentioned in Kim et al. (2024). We

1. **Grammatical Correctness (0-3 points)**:
- 0: Numerous grammatical errors that hinder comprehension.
- 1: Frequent errors that slightly disrupt the reading flow.
- 2: Minor grammatical errors, but the text remains easily readable.
- 3: Fluent and grammatically correct with no noticeable mistakes.
2. **Coherence and Consistency (0-3 points)**:
- 0: The output is disjointed, lacks logical flow, or contradicts itself.
- 1: Some inconsistencies or a lack of clear logical structure.
- 2: Mostly coherent, though minor clarity issues may be present.
- 3: The response is logically consistent and well-organized.
3. **Clarity of Expression (0-2 points)**:
- 0: The output is vague or ambiguous, making the response unclear.
- 1: Generally clear, though some areas may lack precision or clarity.
- 2: Clear, concise, and directly addresses the question or task.
4. **Style and Tone Appropriateness (0-2 points)**:
- 0: Inappropriate tone for the domain (e.g., overly casual or formal for the task).
- 1: Generally appropriate tone, but occasional mismatches with the task context.
- 2: Consistent tone that is well-suited to the operational context.
5. **Answer Completion (0-2 points)**:
- 0: The response is incomplete or significantly deviates from the expected format.
- 1: Response mostly follows the expected format but misses some details.
- 2: The response fully meets the structural and format requirements of the question.

Figure 2: Scoring rubrics for Fluency metric.

use Qwen2-72B-Instruct as the evaluation model, for its strong performance in general language generation (QwenLM, 2023) and its consistent multilingual capabilities without significant degradation. We assess the fluency of various model outputs, scoring them based on grammar, coherence, clarity, appropriateness of style, and answer completeness, as shown in the Figure 2.

**Accuracy.** Traditional metrics such as BLEU and ROUGE fall short in this vertical domain because they often fail to capture the key factual content within long-form responses. This results in inflated scores due to irrelevant word matches, making these metrics insufficient for accuracy evaluation in the highly specialized and knowledge-driven Ops context. To address these shortcomings, we take

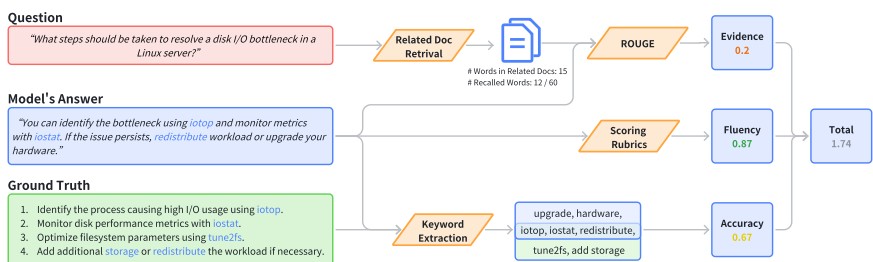

Figure 3: The FAE-Score pipeline.

inspiration from Es et al. (2024), using a keyword extraction method to evaluate the accuracy of model outputs. A judge model (OpenAI, 2023a) is then employed to match the keywords from the model's response with the keywords from the standard answer. The final accuracy score is calculated by determining the F1-Score, which balances precision and recall for the matched keywords.

$$\textbf{Accuracy} = 2 \cdot \frac{\text{P} \cdot \text{R}}{\text{P} + \text{R}}, \quad \text{P} = \frac{\#\text{Matched Keywords}}{\#\text{Keywords in Model Output}}, \quad \text{R} = \frac{\#\text{Matched Keywords}}{\#\text{Keywords in Ground Truth}} \quad (1)$$

**Evidence.** Model responses must not only be accurate but also well-supported by relevant, authoritative information. To evaluate the evidence behind a model's response, we implement a ROUGE-based method to measure the overlap between the generated output and the content of related documents retrieved through similarity search. We used bge-large-zh (Xiao et al., 2023) for document embedding and FAISS (Douze et al., 2024) for similarity search. By retrieving documents that closely match the question, we can assess whether the model's response appropriately references or aligns with this external information. We use ROUGE, as a recall-oriented metric, captures how much of the content in the relevant documents is reflected in the model's output. This ensures that the model does not simply generate plausible-sounding answers but grounds its responses in factual evidence from trusted sources.

$$\textbf{Evidence} = \text{ROUGE}_{\text{Recall}}(R, D) = \frac{\#\text{Overlapping Words}}{\#\text{Words in } D} \quad (2)$$

### 3.5 OPEN-SOURCE POLICY

We have released 20% of the OpsEval dataset to the public to foster contributions from the Ops community and support research. To ensure balanced distribution, this subset was randomly sampled from each data source and sub-domain in proportion to their respective weights. Additionally, for questions involving proprietary company data, we carefully reviewed and modified the content to remove any sensitive information. This sample dataset provides researchers with insights into the types and topics of questions expected in the benchmarks, allowing them to better understand the scope of the evaluation. The sampled dataset also enables model developers to conduct local evaluations of their models, facilitating faster iterations. Moreover, this dataset can serve as a seed for generating QA pairs through automatic QA generation algorithms (Wang et al., 2023c), contributing to the growth of Ops-specific data for future model development. While this subset is available for users' self-evaluation, the complete dataset remains undisclosed. By ensuring that the test set answers are not leaked, we guarantee the reliability and non-leakage of the OpsEval benchmark.

## 4 RESULT ANALYSIS

### 4.1 OVERALL PERFORMANCE

The results of the few-shot evaluation with four settings on the Wired Network Operation test set are shown in Figure 4. Results of the other sub-domains and settings are shown in Appendix C.4. While closed source models like GPT-4 and Claude-3-Opus performs well on the OpsEval benchmark, open-sourced LLMs yield generally worse evaluation results than those in general domains

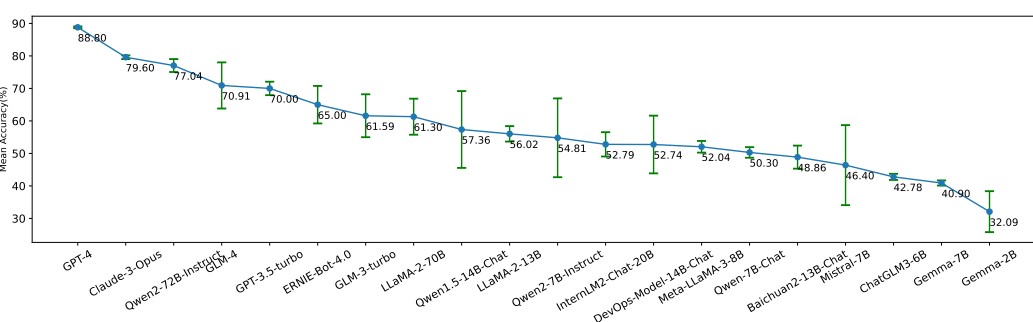

Figure 4: LLMs' overall performance on Wired Network Operations English test set (3-shot). Models are ranked based on their mean accuracy among different settings. The error bars represent the variance in the model's accuracy across different prompting techniques.

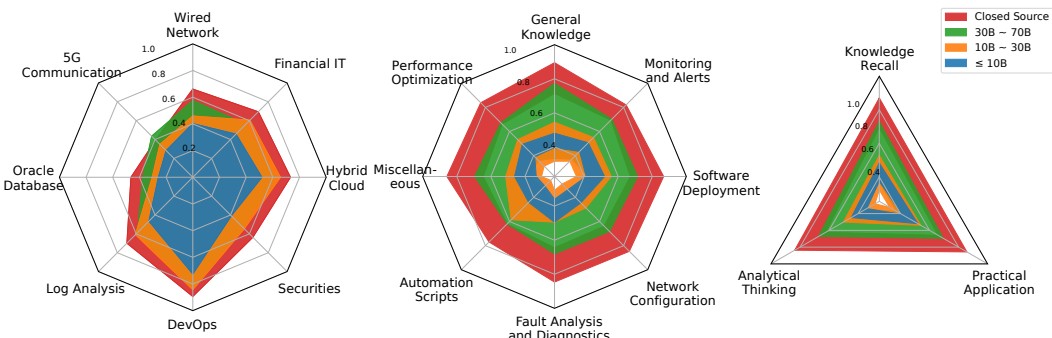

Figure 5: LLMs' performance on eight Ops sub-domains, eight tasks and three abilities. Each colored area presents the lower and upper bound of the corresponding parameter-size group.

like MMLU (Hendrycks et al., 2021) and CEval (Huang et al., 2023). This comparison highlights the necessity of explicitly fine-tuning OpsLLM for the Ops field. Recent open-sourced models like Qwen2-72B-Chat, exhibit competitive performance in multi-choice questions, thanks to their fine-tuning process and the quality of their training data. Furthermore, we observed significant variability in how different LLMs respond to various prompt engineering techniques. Given the critical importance of stability in the Ops domain, it is essential to consider a model's sensitivity to prompts when selecting foundation model. Further research into prompt engineering is needed to improve model performance and reliability in this domain.

**Observations:** 1) Few-shot and CoT can significantly increase performance if the model is tuned to adapt to these techniques, while SC may have little influence on highly consistent LLMs. 2) Smaller models with weaker natural language abilities are less stable with advanced prompts. Simpler prompts work better for them.

**Pratical Lesson**: The choice of fundamental models should be a balance between their performance (average score) and robustness (variance) under different prompt settings.

## 4.2 PERFORMANCE ON DIFFERENT TASKS AND ABILITIES

To investigate how LLMs perform in each Ops sub-domain and each task, and to what extent they possess the general abilities, we summarize the result of different parameter-size groups of LLM and plot them on three radar charts in Figure 5. Regarding the eight tasks we tested, LLMs yield higher accuracy in General Knowledge tasks, while their performance drops and varies drastically in highly specialized tasks like Automation Scripts and Network Configuration, reflecting the impact of specialized corpus and domain knowledge on the performance of LLMs. By grouping LLMs by their parameter size, we find that while LLMs with 10B-30B parameters have higher accuracy in their best cases compared with LLMs with no more than 10B parameters, different 10B-20B LLMs' performance varies drastically. To provide systematic practical lessons for researchers in the operations domain on pre-training and fine-tuning OpsLLM, we have analyzed the error rates of LLMs across the 8 tasks and 3 abilities in Figure 6. By examining the focus areas across different categories, we have identified key research targets for capability training.

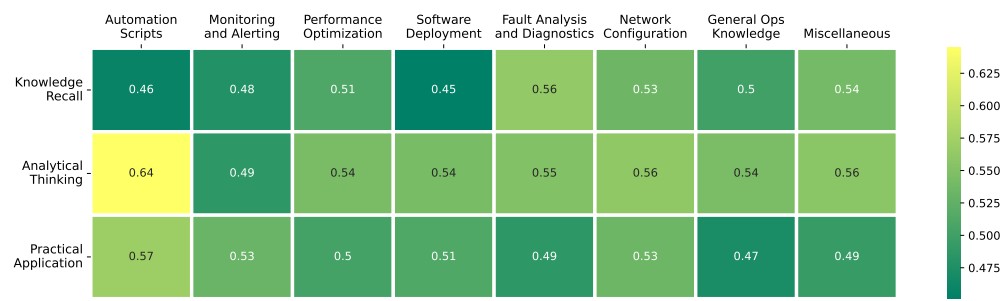

Figure 6: Heatmap of failure case distribution regarding tasks and abilities. The values represent the proportion of failure cases across all LLMs; yellower areas indicate higher failure rates.

Table 3: LLMs' performance on English network operations question-answering problems.

| Model | ROUGE(%) | BLEU(%) | RAGAS(0-10) | Fluency | | Accuracy | | Evidence | | FAE-Total | |
|---|---|---|---|---|---|---|---|---|---|---|---|
| | | | | FAE | Expert | FAE | Expert | FAE | Expert | FAE | Expert |
| GPT-3.5-turbo | 12.26 | 6.78 | 9.23 | 9.38 | 9.12 | 8.06 | 9.65 | 6.21 | 8.11 | 23.65 | 26.88 |
| LLaMA-2-70B | 7.74 | 4.2 | 6.04 | 8.69 | 8.25 | 7.71 | 8.79 | 9.08 | 8.98 | 25.48 | 26.02 |
| LLaMA-2-13B | 4.98 | 3.43 | 8.23 | 8.47 | 9.84 | 7.32 | 9.34 | 8.81 | 7.27 | 24.60 | 26.44 |
| Chinese-Alpaca-2-13B | 3.25 | 1.85 | 5.32 | 5.53 | 8.05 | 6.99 | 7.95 | 6.23 | 6.23 | 18.75 | 22.24 |
| Baichuan-13B-Chat | 4.76 | 0.35 | 7.93 | 7.16 | 7.98 | 8.71 | 7.84 | 6.66 | 7.31 | 22.53 | 23.13 |
| Qwen-7B-Chat | 11.82 | 4.33 | 4.92 | 7.63 | 5.82 | 6.42 | 7.27 | 6.57 | 5.37 | 20.62 | 18.47 |
| ChatGLM2-6B | 9.71 | 5.07 | 5.32 | 5.12 | 7.96 | 6.41 | 6.39 | 6.14 | 4.32 | 17.67 | 18.67 |
| InternLM-7B | 13.27 | 0.54 | 6.21 | 4.99 | 5.16 | 5.00 | 4.90 | 4.75 | 4.28 | 14.74 | 15.77 |
| Chinese-LLaMA-2-13B | 9.19 | 0.24 | 7.34 | 6.98 | 4.64 | 5.29 | 6.32 | 4.63 | 8.34 | 16.90 | 17.88 |

**Observations:** Among the 24 categories of results, models performed the worst in Analytical Thinking for Automation Scripts. This indicates that current models can only recall the learned scripts but struggle to infer their logical relationships. Similarly, Analytical Thinking showed the lowest performance across the three major tasks, indicating that current OpsLLM models still have some way to go before becoming foundational models for Ops Agents. Thus, researchers should focus on inference-related SFT (supervised fine-tuning) datasets.

**Insights:** 1) Among different sub-domains of Ops, 5G communication and database demand further pretraining and fine-tuning of LLM. 2) To be capable of an Ops agent, the foundation model must be able to make a connection between specialized domain knowledge.

## 4.3 PERFORMANCE ON QUESTION-ANSWERING

Table 3 presents the evaluation results of 200 question-answering English questions across four metrics: ROUGE, BLEU, RAGAS, FAE-Score, and Expert-Evaluation. To gain more insight into how different metrics perform in QA evaluation, we use Figure 20 (see in Appendix C.9.2) as a case analysis. While BLEU and ROUGE are efficient in natural language comparison, they lack semantic information to determine which part of the context is more important than others. Knowing that a given benchmark evaluates QA based on BLEU/ROUGE, there is an obvious way to trick the metric: repeat patterns occurring in the question, gaining a higher possibility to match some patterns in the reference answer. Due to their lack of semantic information related to Ops and the potential hack, traditional metrics like BLEU are unsuitable for specialized benchmarks. Instead, with specialized prompting and seperately designed methodology for each criterion (Fluency, Accuracy and Evidence), FAE-Score can comprehensively evaluate models' QA performance, with the Accuracy metric picking up those important keywords and not be influenced by repeated words that contain no useful information, and the Evidence metric checking the recall of relevant supporting contents. In Section 5, we discuss the alignment between different metrics and expert evaluation, validating the effectiveness of FAE-Score in automated QA evaluation within the Ops domain.

**Insight:** In specialized domains, Ops specifically, traditional NLP metrics like BLEU and ROUGE cannot comprehend the key components in the reference answer, resulting their evaluation lacking practical significance. FAE-Score is suitable for large-scale qualitative evaluations in the Ops field.

Table 4: Validation results.

(a) Measurement of potential test data leakage during the training of LLM.

| Dataset | $L_{test}$ | $L_{ref}$ | $\Delta L$ | $\geq 0$? |
|---|---|---|---|---|
| Alpaca | 1.9940 | 2.3542 | -0.3602 | ✗ |
| Alpaca-GPT4 | 1.4988 | 1.7636 | -0.3910 | ✗ |
| CEval | 2.5708 | 2.3099 | 0.2608 | ✓ |
| MMLU | 2.5475 | 2.1898 | 0.3577 | ✓ |
| OpsEval | 2.9854 | 2.6280 | 0.3050 | ✓ |

(b) Pearson correlation coefficients between Expert-Evaluation metrics and Automated metrics. Total is the sum of Fluency, Accuracy, and Evidence.

| Metric | Total | Flu. | Acc. | Evi. |
|---|---|---|---|---|
| ROUGE | -0.44734 | -0.49207 | -0.40889 | -0.31821 |
| BLEU | 0.47139 | 0.46369 | 0.55330 | 0.05977 |
| RAGAS | 0.57169 | 0.40029 | 0.51151 | 0.41928 |
| FAE-Score | **0.91848** | **0.54757** | **0.81523** | **0.58160** |

### 4.4 PERFORMANCE ON DIFFERENT QUANTIZATION PARAMETERS

We conducted experiments on different quantized versions of LLaMA-2-70B and obtained various results and conclusions. For detailed results, please see Appendix C.5. Overall, although the performance of the INT4 version decreases in both English and Chinese, the decline does not exceed 10%. However, the performance drop in the INT3 version is more significant, requiring careful consideration in practical applications.

**Practical Lesson:** Quantization with more than 3 bits can effectively reduce computation and memory costs while preserving performance.

## 5 VALIDATION

### 5.1 BENCHMARK LEAKAGE TEST

For the fairness of a benchmark suited for LLM, avoiding potential bias emerging from test set leakage is necessary. We adapted the methodology from Wei & et.al. (2023) to perform a leakage test on OpsEval's dataset. We evaluate the LLM loss on samples from different datasets for several LLMs and calculate the average loss. For each dataset, we compare LLM loss on the test split ($L_{test}$) and a specially curated reference set ($L_{ref}$) generated by GPT-4, designed to mimic the testing dataset. While Wei & et.al. (2023) only asked GPT-4 to generate similar questions to the GSM8K (Cobbe et al., 2021) dataset, we require GPT-4 to rewrite the question while preserving its original meaning and accuracy.[2] We define a key metric: $\Delta L = L_{test} - L_{ref}$, with a threshold of $\Delta L < 0$ indicating potential test data leakage. A negative $\Delta L$ suggests that the LLM's lower $L_{test}$ comes from overfitting the test set rather than understanding the questions, indicating potential leakage. Table 4a shows the results of leakage measurement. In addition to the two standard evaluation benchmarks (CEval (Huang et al., 2023) and MMLU (Hendrycks et al., 2021)), we conducted the same experiments on the alpaca dataset (Taori et al., 2023) and the Alpaca-GPT4 dataset (Peng et al., 2023), which is likely used in the pre-training of large models, using its $\Delta L$ as reference. This demonstrates the unbiased nature and non-leakage of the OpsEval test set. The models used in the leakage test are listed in Appendix C.1.

### 5.2 EXPERT ALIGNMENT OF FAE-SCORE

Table 4b shows the correlation coefficients between various automated scoring metrics (ROUGE, BLEU, RAGAS, and FAE-Score) and Expert-Evaluation criteria. The results indicate that ROUGE and BLEU scores often misalign with Expert-Evaluation. This misalignment occurs because LLMs with poor performance may generate keywords that boost ROUGE and BLEU scores, while stronger LLMs might receive lower scores due to different wording from standard answers. While RAGAS (Es et al., 2024) aligns better with experts than ROUGE and BLEU, there is still a gap between its scoring rankings for different models and expert judgement standards. In contrast, FAE-Score rankings closely match Expert-Evaluation, particularly with the Accuracy metric. This suggests that FAE-Score is more reliable in assessing the factual accuracy of LLMs' outputs. Notably, GPT-4's performance in factual accuracy is reflected in its strong alignment with the Accuracy metric.

---

[2]For a case example, please see Appendix C.8

## 6 DISCUSSION

### 6.1 AUTOMATED QA GENERATION

During the data collection process, we explored automating question-answer generation. Initially, we sampled QA pairs and manually evaluated their accuracy and domain relevance. Later, we utilized representative examples for few-shot learning, enabling GPT to generate and evaluate QA pairs automatically based on predefined criteria.

Recognizing that most existing benchmarks focus primarily on simple knowledge-based questions, we designed various task-specific templates to address this limitation. These templates require the model to complete specific fields within the template using the provided knowledge content, rather than generating entire questions and answers. This prompt engineering approach allows us to generate detailed and context-specific Ops tasks based on extensive operational knowledge while improving the model's instruction-following ability. By focusing on field-level completion, the overall structure of the QA remains consistent and accurate. In the appendix, we provide the prompt template used for automatic QA generation (Figure 11), along with some task cases illustrating their application (Figure 12). This approach ensures a more diverse and comprehensive evaluation of model capabilities while maintaining the relevance and quality of generated tasks.

### 6.2 FUTURE WORK

**Comprehensive Error Analysis.** To better understand the limitations of large models in Ops question answering, we will further look into the failure cases and identify common error modes, including lack of domain knowledge, hallucinations, inaccurate reasoning, and overconfidence in incorrect answers. We believe this detailed error analysis provides a clearer picture of the challenges faced by models and informs future research directions to address these issues. **Dataset Scale and Real-World Data.** While privacy constraints limit real-world company data, our ongoing collaborations aim to expand the dataset with practical scenarios. Expanding the dataset with real-world scenarios remains a key focus, while the benchmark prioritizes robust evaluation over dataset scale. **Agent and RAG Introduction**: The inclusion of agents and Retrieval-Augmented Generation (RAG) techniques is constrained by the current large models' lack of foundational knowledge in operations. Our leaderboard will incorporate more complex tasks once open-source models possess sufficient operational capabilities.

## 7 CONCLUSION

In this paper, we introduced **OpsEval**, the first comprehensive Ops benchmark suite designed for evaluating the performance of large language models (LLMs) in IT operations. We established a robust evaluation framework encompassing a wide range of sub-domains and tasks within Ops through rigorous data collection from multiple sources and meticulous preprocessing steps. Our benchmark includes a carefully selected set of 9,070 questions, which we have partially released to aid initial evaluations while protecting the integrity of the remaining dataset. It has undergone experiments in data leakage detection, ensuring its reliability. Our observations, supported by quantitative and qualitative results, highlight the need for a balanced approach to selecting fundamental models, considering both performance and robustness. During the QA evaluation, the FAE-Score emerges as a more reliable metric than traditional metrics, suggesting its potential as a replacement for manual labeling in large-scale quantitative evaluations. Our failure rate analysis across 8 tasks and 3 abilities provides researchers with crucial insights and prospects for future breakthroughs. The identified flexibility within the OpsEval framework presents opportunities for future exploration. This benchmark's adaptability facilitates the seamless integration of additional fine-grained tasks, providing a foundation for continued research and optimization of LLMs tailored for Ops.

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

## A DETAILS OF OPSEVAL BENCHMARK

### A.1 INFORMATION ON THE COMPANIES AND EXPERTS PARTICIPATING IN OPSEVAL

Table 5: Information of companies collaborating in OpsEval

| Organization | Domain | URL |
|---|---|---|
| Bank of Shanghai | Financial IT | https://www.bosc.cn/zh/ |
| Bizseer | Ops service/tool provider | https://www.bizseer.com/ |
| ChinaEtek | Internet | https://www.ce-service.com.cn/ |
| Data Foundation | Internet | https://www.dfcdata.com.cn/ |
| Guotai Junan | Securities | https://www.gtja.com/ |
| Huawei | Communication | https://www.huawei.com/ |
| Lenovo | Hybrid Cloud | https://www.lenovo.com/ |
| Rizhiyi | Log Analysis | https://www.rizhiyi.com/ |
| ZTE | Communication | https://www.zte.com.cn/china/ |
| Zabbix | Ops service/tool provider | https://www.zabbix.com/ |
| Inspur | Ops service/tool provider | https://www.inspur.com/ |
| Total | 11 | |

Table 5 shows the companies participating in the creation of OpsEval benchmark suite. Their industries include the Internet, telecommunications, cloud computing, finance, and securities, and each company has dispatched at least two experts to participate in the OpsEval work.

### A.2 DEPENDANCE FILTERING KEYWORD LIST

question_keywords = ['the figure', 'the scenario', 'the previous question']
fail_pred_keywords = ['unclear', 'scenario is not provided', 'cannot be determined', 'none of the options', 'none of the given options']

### A.3 PROMPT FOR GPT-4 CATEGORIZATION

Figure 7 shows the prompt for GPT-4 initial categorization.

### A.4 TASK TYPES OF QUESTIONS

We categorize all questions in OpsEval into 8 tasks. The details of each task are as follows:

- *General Knowledge* pertains to foundational concepts and universal practices within the Ops domain.
- *Fault Analysis and Diagnostics* focuses on detecting and addressing discrepancies or faults within a network or system, and deducing the primary causes behind those disruptions.

> I need your help in analyzing a multi-choice question, determine the domain and the task type it belongs to.
> **Domains:** When classifying the domain, be specific, dive deeper into domains such as: Database/Network Operations
> **Task Types:** For the task type, consider categories like: Monitoring and Alerts, Performance Optimization
> **Summary your response as JSON format: {"domain": "specific_domain", "task": "specific_task_type"}**

Figure 7: The prompt for GPT-4 initial categorization

- *Network Configuration* revolves around suggesting optimal configurations for network devices like routers, switches, and firewalls to ensure their efficient and secure operations.

- *Software Deployment* deals with the dissemination and management of software applications throughout the network or system, verifying their correct installation.

- *Monitoring and Alerts* harnesses monitoring tools to supervise network and system efficiency and implements alert mechanisms to notify administrators of emerging issues.

- *Performance Optimization* is centered on refining the network and system for peak performance and recognizing potential enhancement areas.

- *Automation Scripts* involves the formulation of automation scripts to facilitate processes and decrease manual intervention for administrators.

- *Miscellaneous* comprises tasks that do not strictly adhere to the aforementioned classifications or involve a combination of various tasks.

### A.5 ABILITY LEVELS OF QUESTIONS

Different questions require different levels of ability to answer. We classify all questions in OpsEval into 3 categories. The details of each ability are as follows:

1. *Knowledge Recall:* Questions under this category primarily test a model's capacity to recognize and recall core concepts and foundational knowledge. Such questions are akin to situations where a professional might need to identify a standard procedure or recognize a well-known issue based solely on previous knowledge.
2. *Analytical thinking:* These questions demand more than mere recall. They necessitate a deeper level of thought, expecting the model to dissect a problem, correlate diverse pieces of information, and derive a coherent conclusion. It mirrors real-world scenarios where professionals troubleshoot complex issues by connecting various dots and leveraging their comprehensive understanding.
3. *Practical Application:* These questions challenge a model's ability to apply its foundational knowledge or analytical conclusions to provide actionable recommendations for specific scenarios. It epitomizes situations where professionals are expected to make decisions or suggest solutions based on in-depth analysis and expertise.

Figure 8 illustrates examples in our question set, shedding light on our classification methodology.

### A.6 PROMPT AND FORMATTING OF QUESTIONS

Figure 8 illustrates examples of the questions after our preprocessing pipeline.

### A.7 AN EXAMPLE OF SUBJECTIVE QUESTIONS

A saved subjective question in OpsEval is presented in Figure 9, which contains not only the raw question but also its type of task.

As shown in Figure 10, we combine the task and ability of each question with the question itself as the prompt for LLMs.

Which of the following represents quantifying data moved from one host to another within a specific time frame?
A: Reliability                          B: Response time
C: Throughput                          D: Jitter
Answer: C
Analysis: Throughput is the measure of data transferred from one host to another in a given amount of time
Task: Performance Optimization
Ability: Knowledge Recall

Which command enables a router to signal clients that they should acquire additional configuration details from a DHCPv6 server?
A: ipv6 nd ra suppress                 B: ipv6 dhcp relay destination
C: ipv6 address autoconfig             D: ipv6 nd other-config-flag
Answer: D
Analysis: The **ipv6** nd other-config-flag** command is used to enable a router to inform clients that they need to get additional configuration information from a DHCPv6 server
Task: Automation Scripts
Ability: Analytical Thinking

Question: You receive a call from a user experiencing difficulties connecting to a new VPN. What is the initial step you should take?
A: Find out what has changed.          B: Reboot the workstation.
C. Document the solution.              D: Identify the symptoms and potential causes.
Answer: D
Analysis: Since this is a new connection, you need to start by troubleshooting and identify the symptoms and potential causes
Task: Fault Analysis and Diagnostics
Ability: Practical Application

Figure 8: Three examples of the processed questions

Question: You have a router interface with an IP address of 192.168.192.10/29. What is the broadcast address that the hosts on this LAN will utilize?
问题：路由器上有一个接口，IP地址为192.168.192.10/29。主机在这个局域网上使用的广播地址是什么？

Keypoint: 192.168.192.15
答案要点：192.168.192.15

Detailed Answer: A /29 (255.255.255.248) has a block size of 8 in the fourth octet. This means the subnets are 0, 8, 16, 24, and so on. 10 is in the 8 subnet. The next subnet is 16, so 15 is the broadcast address.
答案解析：/29（255.255.255.248）在第四个八位组有8个块大小。这意味着子网是0，8，16，24等等。10在8的子网中。下一个子网是16，所以15是广播地址。

Task: Network Configuration
任务：网络配置

Ability: Analytical Thinking
能力：推理

Figure 9: An example of the saved subjective questions

A subjective question in OpsEval

Question: You have a router interface with an IP address of 192.168.192.10/29. What is the broadcast address that the hosts on this LAN will utilize?
问题：路由器上有一个接口，IP地址为192.168.192.10/29。主机在这个局域网上使用的广播地址是什么？。

Task: Network Configuration
任务：网络配置

Ability: Analytical Thinking
能力：推理

Prompt

Answer the Reasoning question about Network Configuration.
You have a router interface with an IP address of 192.168.192.10/29. What is the broadcast address that the hosts on this LAN will utilize?
回答关于网络配置的推理问题。
路由器上有一个接口，IP地址为192.168.192.10/29。主机在这个局域网上使用的广播地址是什么？

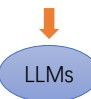

LLMs

Figure 10: An example of building the prompt of subjective questions.

> You are an operations expert, and your task is to generate a question that adheres to the given template or follows a similar format, based on the provided knowledge content.
>
> The question template is as follows:
>
> {question_template}
>
> Here, {{}} represents parameters that need to be filled.
>
> The knowledge points for generating the question are as follows:
>
> {context}
>
> When generating the question, you need to construct a business or operations scenario based on the knowledge content, and then use that scenario to populate the required fields.
>
> When creating the question, you need to adhere to the following constraints:
> {constraints}
>
> Return a JSON object containing the required fields from the template, as well as an answer field and an explanation field. The answer field should contain the answer to the question, and the explanation field should provide reasoning for the answer, explaining why it is correct. The terms used in your question and answer must match the given knowledge content, and you should not invent new terminology.
>
> The reference format is as follows:
>
> {json_example}
>
> Your returned content should not start with ```json. Return the JSON object directly.

Figure 11: Prompt template for automated QA generation

## B  AUTOMATED QA GENERATION

Figure 11 shows the prompt template we used for automated QA generation experiment. Figure 12 shows some automatically generated QAs, their task description, template and example question.

## C  ADDITIONAL DETAILS OF EXPERIMENTS

### C.1  DETAILED INFORMATION OF LLMS EVALUATED

GPT-4 (OpenAI, 2023a) is a large multimodal model (accepting image and text inputs, emitting text outputs) that, while less capable than humans in many real-world scenarios, exhibits human-level performance on various professional and academic benchmarks. It is recognized as the strongest lanuage model currently. ChatGPT (OpenAI, 2022) is an earlier AI-powered language model developed by OpenAI which is built upon GPT-3.5. We use the GPT-3.5-turbo version in our experiments. LLaMA 2 (Touvron & et.al., 2023) is a second-generation open-source LLM from Meta which is very popular due to its open-source feature. It has the ability to process multiple languages including Chinese. We evaluate three weights (70B, 13B and 7B as shown in 6) of LLaMA 2.

Although LLaMA 2 is able to process Chinese input, it has a small Chinese vocabulary so that its abitilty of understanding and generating Chinese text is limited. As a result, we evaluate some Chinese-oriented LLMs which are published by institutions in China. ERNIE-Bot 4.0 (Baidu, 2024) is the latest self-developed language model released by Baidu. As claimed by Baidu, ERNIE-Bot 4.0 rivals OpenAI's GPT-4. Qwen (QwenLM, 2023) (abbr. Tongyi Qianwen) is a series of LLMs developed by Alibaba Cloud. And Qwen-Chat is a series of large-model-based AI assistant trained with alignment techniques based on the pretrained Qwen. We evaluate three weights (72B, 14B and 7B as shown in 6) of Qwen-Chat. Baichuan2-13B-Chat (Baichuan, 2023) is aligned chat model based

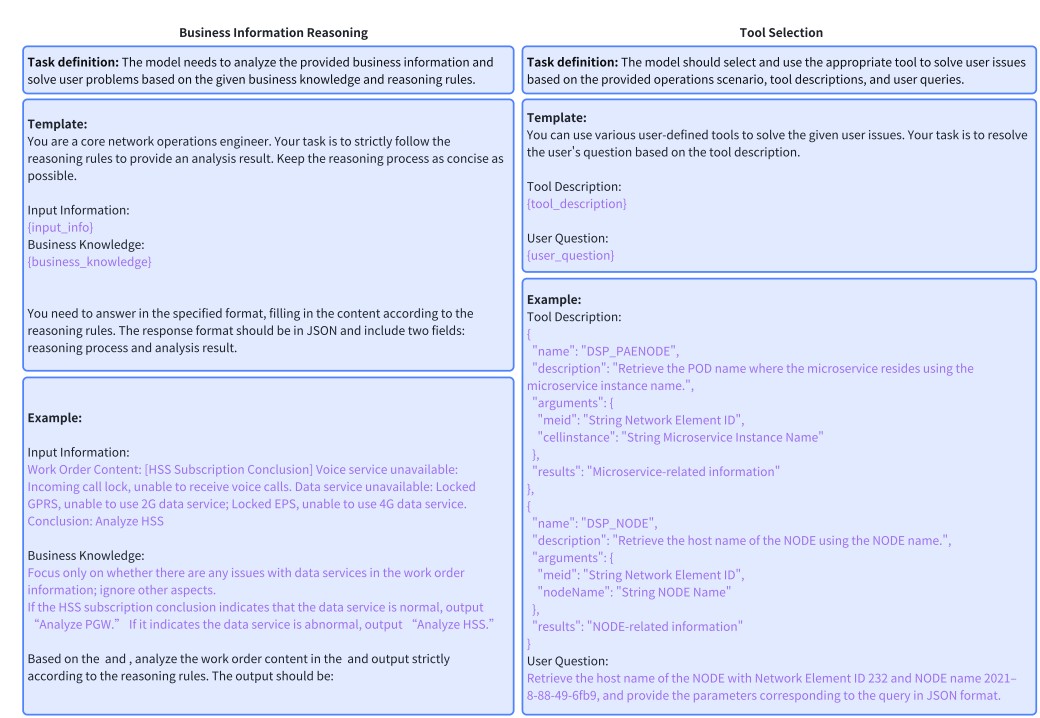

Figure 12: Some automatically generated QAs, their task description, template and example question

Table 6: Models evaluated in this paper. The "access" column in the table shows whether we have full access to the model weights or can only access them through API.

| Model | Creator | #Parameters | Access | License |
|---|---|---|---|---|
| GPT-4/GPT-3.5-turbo | OpenAI | *undisclosed* | API | Proprietary |
| ERNIE-Bot-4.0 | Baidu | *undisclosed* | API | Proprietary |
| GLM4/GLM3-turbo | Tsinghua Zhipu | *undisclosed* | API | Proprietary |
| Meta-LLaMA-3 | Meta | 8B | Weights | Llama 3 Community |
| LLaMA-2 | Meta | 7/13/70B | Weights | Llama 2 Community |
| Qwen-Chat | Alibaba Cloud | 7/14/72B | Weights | Qianwen LICENSE |
| Qwen1.5-Chat | Alibaba Cloud | 14B | Weights | Qianwen LICENSE |
| InternLM2-Chat | Shanghai AI Laboratory | 7/20B | Weights | Apache-2.0 |
| DevOps-Model-Chat | CodeFuse | 14B | Weights | Apache-2.0 |
| Baichuan2-Chat | Baichuan Intelligence | 13B | Weights | Apache-2.0 |
| ChatGLM3 | Tsinghua Zhipu | 6B | Weights | Apache-2.0 |
| Mistral | Mistral | 7B | Weights | Apache-2.0 |
| Gemma | Google | 2/7B | Weights | Gemma license |
| Claude-3-Opus | Anthropic | *undisclosed* | API | Proprietary |
| Qwen2-Instruct | Alibaba Cloud | 7/72B | Weights | Qianwen LICENSE |

Table 7: GPTQ models for LLaMA-2-70B

| Model | Size | #GPTQ Dataset | Disc |
|---|---|---|---|
| LLaMA-2-70B | 140GB | / | Raw LLaMA-2-70B model. |
| LLaMA-2-70B-Int4 | 35.33GB | wikitext | 4-bit quantization model. |
| LLaMA-2-70B-Int3 | 26.78GB | wikitext | 3-bit quantization model. |

on Baichuan2-13B-Base (Baichuan, 2023) which is an open-source LLM published by Baichuan Intelligence. GLM (Du et al., 2022), developed by Tsinghua Knowledge Engineering Group, is a General Language Model pretrained with an autoregressive blank-filling objective and can be finetuned on various natural language understanding and generation tasks. Based on GLM, Zhipu AI released GLM4 (the newest version of GLM model) (Zeng et al., 2022) and GLM3 (the third version of GLM model). For GLM3, we use GLM3-turbo (Zeng et al., 2022) version and ChatGLM3-6B (Zeng et al., 2022) in our experiments. InternLM2-Chat-20B and InternLM2-Chat-7B (InternLM_Team, 2023), recently developed by Shanghai AI Laboratory, are multi-lingual models based on billions of parameters through multi-stage progressive training on over trillions of tokens. Furthermore, we evaluate DevOps-Model-14B-Chat (AI, 2024a), an open source Chinese DevOps oriented models, mainly dedicated to exerting practical value in the field of DevOps.Gemma (Gemma_Team et al., 2024) is a family of lightweight, state-of-the-art open models based on Gemini technology from Google DeepMind. Trained on up to 6T tokens, Gemma achieves excellent language understanding and reasoning capabilities. We conducted an evaluation of Gemma-2b and Gemma-7b to investigate the effectiveness of Gemma with different weights.

In general, since some models (among them GPT-4, GPT-3.5-turbo, ERNIE-Bot-4.0, GLM4, GLM3-turbo) are not locally available, we evaluate them via API calls. For the remaining models, we perform local inference during evaluation.

## C.2 PROMPTS

Figure 13: An example of zero-shot evaluation in the CoT setting.Black font represents prompts in English. Purple font represents prompts in Chinese. Red font represents the model's output in Chinese. Dark red font represents the model's output in English.

For zero-shot evaluation in the CoT setting, we get the answer of LLMs in two rounds. Firstly, by adding a 'Let's think step by step.' after the question, LLMs will output its reasoning result. Secondly, we compose the final prompt of the question and the reasoning result in whole as the input of LLMs to get the final answer. An example is shown in Figure 13. For few-shot evaluation in the CoT setting, We make an analysis of each option of the question as a reasoning process, and craft three Q-A examples with CoT reasoning process in answers. An example is shown in Figure 14.

Here is a single-answer multiple choice question about Networking Fundamentals.
以下关于网络基础知识的单选选择题，请直接给出正确答案的选项。

Which devices can transmit packets across multiple networks and use tables to store network addresses to determine the optimal destination?
什么设备可以在多个网络之间传输数据包，并使用表格存储网络地址以确定最佳目的地？

A: Hubs    B: Firewalls  C: Routers   D: Switches
A: 集线器   B: 防火墙   C: 路由器   D: 交换机

Answer: A-Hubs⋯⋯, B-Firewalls⋯⋯, C-Routers⋯⋯, D-Switches⋯⋯. So the answer is C.
答：A-集线器⋯⋯，B-防火墙⋯⋯，C-路由器⋯⋯，D-交换机⋯⋯。所以答案是C。

**... [3-shot examples] ...**

Here is a single-answer multiple choice question about Network Implementations.
以下关于网络实现的单选选择题，请直接给出正确答案的选项。

Which TCP/IP routing protocol among the following does not incorporate the subnet mask in its route update messages, thereby hindering its support for subnetting?
以下哪个TCP/IP路由协议在其路由更新消息中不包括子网掩码，从而无法支持子网划分？

A: Routing Information Protocol, version 1 (RIPv1) B: Routing Information Protocol, version 2 (RIPv2)
C: Border Gateway Protocol (BGP)          D: Open Shortest Path First (OSPF)
A: 路由信息协议，版本1（RIPv1） B: 路由信息协议第二版（RIPv2）
C: 边界网关协议（BGP）          D: 开放最短路径优先（OSPF）

Answer: A-Routing Information Protocol⋯⋯. So the answer is A.
答：A-路由信息协议⋯⋯，所以答案是A。

Figure 14: An example of few-shot evaluation in the CoT setting. Black font represents prompts in English. Purple font represents prompts in Chinese. Red font represents the model's output in Chinese. Dark red font represents the model's output in English.

## C.3 COMPUTE AND RESOURCES USED FOR EXPERIMENTS

During our OpEval experiments evaluating different LLMs, we utilize an 8 Nvidia A800-80GB GPU cluster to run inference on models with available weights. For models with API access, we perform inference using CPUs.

## C.4 OVERVIEW PERFORMANCE ON DIFFERENT TEST SETS

Table 8: LLMs' overall performance on wired network operations test set

| Model | English Test Set | | | | | | | | Chinese Test Set | | | | | | | |
|---|---|---|---|---|---|---|---|---|---|---|---|---|---|---|---|---|
| | Zero-shot | | | | 3-shot | | | | Zero-shot | | | | 3-shot | | | |
| | Naive | SC | CoT | CoT+SC | Naive | SC | CoT | CoT+SC | Naive | SC | CoT | CoT+SC | Naive | SC | CoT | CoT+SC |
| GPT-4 | / | / | / | / | / | / | **88.70** | / | / | / | / | / | / | / | **86.00** | / |
| Qwen-72B-Chat | 70.41 | 70.50 | 72.38 | **72.56** | 70.32 | 70.32 | 70.13 | 70.22 | 65.77 | 65.86 | 68.13 | 68.30 | 69.40 | 69.40 | 69.99 | **70.08** |
| GPT-3.5-turbo | 66.60 | 66.80 | 69.60 | 72.00 | 68.30 | 68.30 | 70.90 | **72.50** | 58.40 | 58.60 | 64.80 | **67.60** | 59.20 | 59.70 | 65.20 | 67.40 |
| ERNIE-Bot-4.0 | 61.15 | 61.15 | 70.00 | 70.00 | 60.00 | 60.00 | 70.00 | **70.00** | 67.54 | 67.54 | 71.96 | 71.96 | 72.00 | 72.00 | 78.00 | **78.00** |
| Qwen1.5-14B-Chat | 54.90 | 34.88 | 64.09 | 60.82 | 52.23 | **65.55** | 59.54 | 47.08 | 54.04 | 45.18 | 62.56 | 59.12 | 58.78 | 61.10 | **63.43** | 52.5 |
| Devops-Model-14B-Chat | 30.69 | 30.59 | 55.77 | 63.63 | **63.85** | 61.96 | 41.15 | 44.01 | 47.59 | 46.57 | 52.52 | 56.01 | **62.07** | 60.08 | 50.59 | 55.79 |
| Qwen-14B-Chat | 43.78 | 47.81 | 56.58 | 59.40 | **62.09** | 59.70 | 49.06 | 55.88 | 48.35 | 48.81 | 55.35 | 57.40 | **58.53** | 56.12 | 52.12 | 54.99 |
| LLaMA-2-13B | 41.80 | 46.50 | 53.10 | 58.70 | 53.30 | 53.60 | 51.60 | **61.00** | 29.70 | 31.60 | 51.60 | **57.00** | 39.60 | 38.90 | 48.00 | 50.60 |
| Gemma-7B | 25.09 | 25.09 | 50.86 | 50.86 | **59.12** | **59.12** | 50.77 | 50.77 | 31.58 | 31.58 | 47.59 | 47.59 | 34.68 | 34.68 | **48.88** | **48.88** |
| LLaMA-2-70B-Chat | 25.29 | 25.29 | 57.97 | 58.06 | 52.97 | 52.97 | **58.55** | **58.55** | 38.55 | 38.55 | **57.49** | **57.49** | 49.09 | 49.09 | 48.57 | 48.57 |
| Internlm2-Chat-20B | **56.36** | **56.36** | 26.18 | 26.18 | 60.48 | 60.48 | 45.10 | 45.10 | 57.49 | 57.49 | 57.14 | 57.14 | **59.12** | **59.12** | 50.77 | 50.77 |
| Internlm2-Chat-7B | 49.74 | 49.74 | **56.19** | **56.19** | 48.20 | 48.20 | 49.74 | 49.74 | 57.49 | 57.49 | 57.14 | 57.14 | **59.12** | **59.12** | 50.77 | 50.77 |
| LLaMA-2-7B | 39.50 | 40.00 | 45.40 | 49.50 | 48.20 | 46.80 | 52.00 | **55.20** | 29.80 | 30.20 | 50.10 | **55.60** | 38.60 | 40.80 | 45.60 | 50.40 |
| Qwen-7B-Chat | 45.90 | 46.00 | 47.30 | 50.10 | **52.10** | 51.00 | 48.30 | 49.80 | 29.60 | 29.90 | 50.60 | **53.50** | 50.40 | 46.90 | 46.90 | 47.70 |
| Baichuan2-13B-Chat | 37.90 | 38.30 | 42.70 | 46.60 | **51.90** | 51.60 | 44.50 | 47.45 | 44.60 | 45.40 | 41.60 | 44.30 | 45.60 | 45.70 | 43.90 | **46.70** |

Note: The best accuracy of each language for each LLM is in **bold** font.

In Table 8, Table 9 and Table 10, we present overview performance of different LLMs on the 3 test sets in OpsEval, including Wired Network Operations, 5G Communication Technology Operations and Database Operations.

## C.5 PERFORMANCE ON DIFFERENT QUANTIZATION MODELS

Figure 15 shows the accuracy of LLaMA-2-70B of different quantization parameters on objective questions, English and Chinese questions respectively. We do both zero-shot and few-shot evaluation with the naive setting.

Table 9: LLMs' overall performance on 5G communication operations test set

| Model | English Test Set | | | | | | | | Chinese Test Set | | | | | | | |
|---|---|---|---|---|---|---|---|---|---|---|---|---|---|---|---|---|
| | Zero-shot | | | | 3-shot | | | | Zero-shot | | | | 3-shot | | | |
| | Naive | SC | CoT | CoT+SC | Naive | SC | CoT | CoT+SC | Naive | SC | CoT | CoT+SC | Naive | SC | CoT | CoT+SC |
| GPT-4 | / | / | 56.30 | **65.49** | / | / | 59.62 | 63.54 | / | / | 57.19 | 62.11 | / | / | 61.55 | **65.68** |
| Qwen-72B-Chat | 53.19 | 53.19 | 55.25 | 55.52 | 58.13 | 58.13 | 58.72 | **58.99** | 64.79 | 64.79 | **65.79** | 65.72 | 70.19 | 70.19 | 68.31 | 68.38 |
| InternLM2-Chat-20B | 39.10 | 39.10 | 37.70 | 37.70 | **47.70** | **47.70** | 33.50 | 33.50 | 44.60 | 44.60 | 47.00 | 47.00 | **62.20** | **62.20** | 38.30 | 38.30 |
| Qwen-14B-Chat | 33.71 | 36.25 | 41.24 | 42.51 | 51.19 | 50.39 | 57.18 | **59.18** | 41.71 | 41.44 | 45.58 | 47.98 | 53.52 | 49.92 | 54.72 | **58.85** |
| DevOps-Model-14B-Chat | 31.04 | 30.51 | 42.84 | 47.37 | **52.25** | 49.38 | 45.90 | 47.23 | 41.04 | 42.70 | 48.71 | 53.57 | 56.85 | **57.25** | 51.30 | 54.29 |
| ERNIE-Bot-4.0 | 43.66 | 43.66 | **51.99** | **51.99** | 44.00 | 44.00 | 50.00 | 50.00 | 45.99 | 45.99 | 48.98 | 48.98 | 46.00 | 46.00 | **54.00** | **54.00** |
| LLaMA-2-70B | 23.64 | 23.64 | 39.31 | 39.31 | 38.98 | 39.12 | **47.90** | **47.90** | 24.38 | 24.38 | 43.63 | 43.63 | 44.65 | 44.65 | **48.84** | **48.84** |
| Mistral-7B | 26.91 | 26.91 | 30.65 | 30.65 | 40.52 | 40.52 | **46.84** | **46.84** | 1.27 | 1.27 | 42.05 | 42.05 | 30.72 | 30.72 | **46.44** | **46.44** |
| InternLM2-Chat-7B | 36.80 | 36.80 | 31.70 | 31.70 | **46.30** | **46.30** | 36.90 | 36.90 | 38.80 | 38.80 | 44.60 | 44.60 | **46.00** | **46.00** | 35.80 | 35.80 |
| Gemma-7B | 23.10 | 23.10 | **34.40** | **34.40** | 21.40 | 21.40 | 33.10 | 33.10 | 27.30 | 27.30 | 35.40 | 35.40 | 17.30 | 17.30 | **44.50** | **44.50** |
| LLaMA-2-13B | 15.62 | 18.32 | 29.88 | 34.45 | 23.16 | 29.14 | 37.59 | **44.3** | 25.43 | 27.16 | 29.17 | 29.99 | 36.56 | 36.15 | 37.70 | **39.02** |
| GPT-3.5-turbo | 34.92 | 34.82 | 38.53 | **43.50** | 39.40 | 39.19 | 40.93 | 42.58 | 36.98 | 36.83 | 37.95 | 39.25 | 39.17 | 39.77 | 41.93 | **42.15** |
| Qwen-7B-Chat | 33.85 | 33.74 | 32.45 | 34.10 | 32.91 | 32.70 | **36.65** | **36.65** | 36.27 | 36.50 | 33.27 | 33.51 | **42.22** | 40.59 | 31.28 | 31.46 |
| ChatGLM3-6B | 30.40 | 30.40 | 30.70 | 30.70 | 26.90 | 26.90 | **37.20** | **37.20** | 32.60 | 32.60 | 35.40 | 35.40 | 28.30 | 28.30 | **40.90** | **40.90** |
| Baichuan2-13B-Chat | 14.10 | 15.30 | 24.10 | 25.80 | 32.30 | **33.10** | 25.60 | 27.70 | 35.64 | **35.91** | 30.59 | 30.52 | 34.65 | 35.6 | 30.21 | 32.05 |
| LLaMA-2-7B | 19.14 | 21.62 | 25.70 | 27.11 | 21.38 | 24.85 | 32.38 | **34.83** | 23.57 | 23.47 | 27.65 | 29.26 | 30.30 | 30.03 | 30.98 | **31.93** |
| Gemma-2B | 20.10 | 20.10 | 24.20 | 24.20 | 31.20 | 31.20 | **35.50** | **35.50** | 25.60 | 25.60 | 28.30 | 28.30 | 19.10 | 19.10 | **35.50** | **35.50** |

Note: The best accuracy of each language for each LLM is in **bold** font.

Table 10: LLMs' overall performance on database operations test set

| Model | English Test Set | | | | | | | | Chinese Test Set | | | | | | | |
|---|---|---|---|---|---|---|---|---|---|---|---|---|---|---|---|---|
| | Zero-shot | | | | 3-shot | | | | Zero-shot | | | | 3-shot | | | |
| | Naive | SC | CoT | CoT+SC | Naive | SC | CoT | CoT+SC | Naive | SC | CoT | CoT+SC | Naive | SC | CoT | CoT+SC |
| GPT-4 | / | / | 59.02 | **64.56** | / | / | 58.35 | 62.58 | / | / | 59.38 | **65.17** | / | / | 44.06 | 48.09 |
| InternLM2-Chat-20B | / | / | **59.21** | **59.21** | / | / | / | / | / | / | / | / | / | / | / | / |
| ERNIE-Bot-4.0 | 43.80 | 43.80 | 47.14 | 47.14 | 46.00 | 46.00 | **54.0** | **54.0** | 48.56 | 48.56 | 50.64 | 50.64 | 48.00 | 48.00 | **54.0** | **54.0** |
| Gemma-7B | 14.29 | 14.29 | 30.99 | 30.99 | 2.60 | 2.60 | **43.86** | **43.86** | 19.32 | 19.32 | **53.95** | **53.95** | 18.51 | 18.51 | 5.20 | 5.20 |
| Qwen-72B-Chat | 47.28 | 47.48 | 48.09 | 48.09 | **49.70** | **49.70** | 43.46 | 43.66 | 48.29 | 48.49 | 49.50 | **49.70** | **49.70** | **49.70** | 45.27 | 44.87 |
| GPT-3.5-turbo | 38.63 | 38.83 | 40.04 | 42.05 | 36.62 | 37.63 | 42.66 | **43.86** | 36.42 | 35.81 | 39.24 | **43.26** | 39.84 | 39.44 | 27.16 | 27.77 |
| Qwen-14B-Chat | 24.95 | 28.37 | 33.00 | **36.62** | 27.97 | 28.37 | 27.97 | 24.14 | 27.57 | 27.57 | 32.39 | 36.02 | **40.04** | 35.41 | 30.38 | 33.40 |
| DevOps-Model-14B-Chat | 25.15 | 26.96 | 35.41 | **38.83** | 33.20 | 34.81 | 27.36 | 27.36 | 24.75 | 22.74 | 28.37 | 27.77 | 36.62 | **37.02** | 27.57 | 26.36 |
| LLaMA-2-70B | 19.72 | 19.72 | 27.97 | 27.97 | 26.56 | 26.56 | **32.6** | **32.6** | 15.29 | 15.29 | **34.81** | **34.81** | 26.76 | 26.76 | 33.80 | 33.80 |
| Qwen-7B-Chat | 18.91 | 19.11 | 22.13 | 23.94 | 26.76 | 25.55 | **34.81** | **34.81** | 18.51 | 17.71 | 27.36 | 28.37 | 29.78 | 29.58 | **33.60** | **33.60** |
| LLaMA-2-13B | 16.10 | 20.32 | 23.94 | 29.58 | 20.12 | 22.33 | 24.35 | **33.80** | 23.94 | 24.35 | 29.58 | **31.99** | 24.55 | 26.76 | 21.13 | 20.72 |
| LLaMA-2-7B | 22.13 | 23.74 | 23.74 | 26.56 | 19.32 | 20.52 | 28.77 | **33.60** | 20.72 | 20.72 | 27.16 | **27.97** | 21.53 | 18.51 | 18.31 | 17.91 |
| Mistral-7B | 17.10 | 17.10 | 26.76 | 26.76 | **31.19** | **31.19** | 27.97 | 27.97 | 0.20 | 0.20 | 26.76 | 26.76 | 10.26 | 10.26 | **32.19** | **32.19** |
| InternLM2-Chat-7B | 27.16 | 27.16 | 28.17 | 28.17 | 29.98 | 29.98 | **30.18** | **30.18** | 28.57 | 28.57 | **31.79** | **31.79** | 30.78 | 30.78 | 31.19 | 31.19 |
| ChatGLM3-6B | 20.93 | 20.93 | 25.15 | 25.15 | 24.75 | 24.75 | **29.18** | **29.18** | 21.33 | 21.33 | 28.97 | 28.97 | 21.73 | 21.73 | **29.58** | **29.58** |
| Baichuan2-13B-Chat | 17.10 | 19.11 | 18.71 | 22.94 | 25.96 | **26.56** | 20.93 | 24.55 | 25.75 | 25.55 | 20.12 | 21.33 | **27.77** | 26.76 | 22.74 | 24.75 |
| Gemma-2B | 16.90 | 16.90 | 19.52 | 19.52 | 16.10 | 16.10 | **24.75** | **24.75** | 18.51 | 18.51 | 24.95 | 24.95 | 21.53 | 21.53 | **27.77** | **27.77** |

Note: The best accuracy of each language for each LLM is in **bold** font.

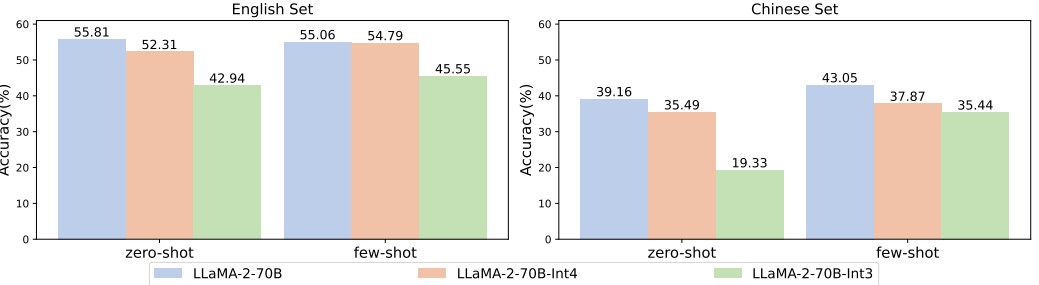

Figure 15: LLaMA-2-70B's performance of different quantization parameters. Both zero-shot and few-shot evaluations have been conducted on Wired Network Operations test set under the naive setting.

LLaMA2-70B-Int4 can achieve an accuracy close to LLaMA-2-70B without quantization. Specifically, on English multi-choice questions, the accuracy of the GPTQ model with 4-bit quantization parameters is 3.50% lower in zero-shot evaluation and 0.27% in few-shot evaluation compared to LLaMA-2-70B. As for Chinese questions, the accuracy of LLaMA2-70B-Int4 is 3.67% lower in zero-shot evaluation and 5.18% in few-shot evaluation compared to LLaMA-2-70B. However, LLaMA2-70B-Int3 has a performance degradation that cannot be ignored. On average, the accuracy of LLaMA2-70B-Int3 in English set has a 12.46% degradation compared to LLaMA-2-70B and a 9.30% degradation compared to LLaMA2-70B-Int4.

## C.6 Performance on Different Languages

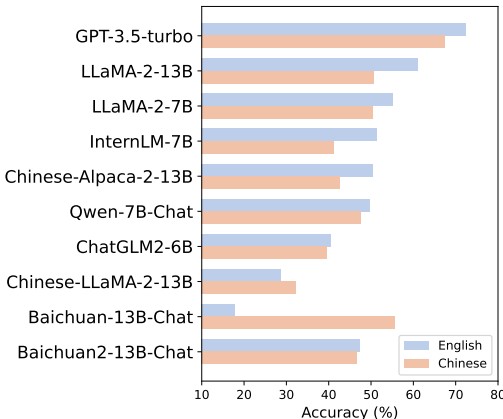

Figure 16: LLMs' few-shot performance on English/Chinese test set (CoT+SC)

In Figure 16, we compare the few-shot performance of various LLMs under the CoT+SC setting for both English and Chinese questions. Notably, some of the LLMs that have undergone specific training or fine-tuning with Chinese language corpus, such as Chinese-Alpaca-2-13B, Qwen-7B-Chat, and ChatGLM2-6B, still perform better in answering English questions than Chinese ones.

Despite the observed fact that performance tends to be lower for Chinese questions compared to the original English questions, we can still glean valuable insights into the language capabilities of the LLMs. Notably:

1. ChatGLM2-6B experiences the smallest decline in performance when transitioning to Chinese questions. *This improvement can be attributed to its substantial exposure to Chinese language data during training rather than simple fine-tuning on top of an existing base model.*
2. LLaMA-2-13B exhibits the most significant drop in performance when switching to Chinese questions. *This indicates that the shift in language impacts LLMs' general understanding ability and capacity to extract domain-specific knowledge.*

We also observe an interesting phenomenon with the Baichuan-13B-Chat in the 3-shot evaluation with the CoT+SC setting, where its performance in Chinese questions significantly outperforms in English. We examine the LLM's outputs and analyze a sample question to shed light on this phenomenon in Appendix C.9.4.

## C.7 Expert alignment of FAE-Score

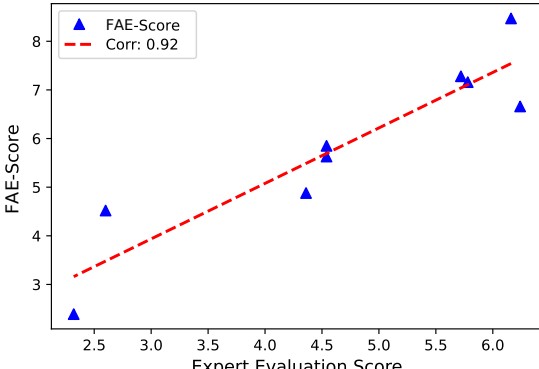

Figure 17: Scatter plot and trendline of FAE-Score compared to Expert Evaluation score.

As depicted in Figure 17, the FAE-Score demonstrates a strong positive correlation with Expert Evaluation Score, making it a valuable and effective substitute for automated evaluation.

## C.8 LEAKAGE TEST EXAMPLE

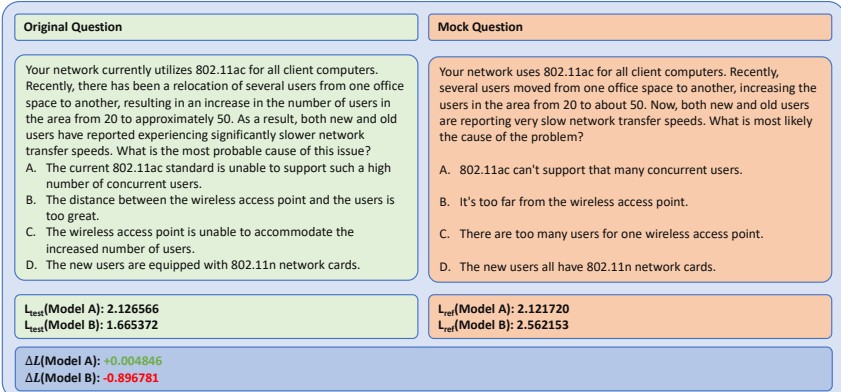

Figure 18: An example for leakage Test.

Figure 18 shows an example for leakage test. Note that the mock question is a paraphrase of the original question, but uses different vocabulary and phrasing. The difference in $\Delta L$ for Model B suggests that there may be potential leakage in the original question.

## C.9 CASE STUDY

### C.9.1 FAILURE CASES OF OPSEVAL QUESTIONS

Figure 19 illustrates three instances of failure in certain LLMs. The primary reasons for these failures are typically attributed to either a deficiency in domain knowledge or a lack of comprehension of domain knowledge.

### C.9.2 CASE ANALYSIS ON QA METRICS

Figure 20 illustrates an example where BLEU/ROUGE fail to pick up important keywords and are influenced by the repeated pattern attack, while FAE-Score avoids these two issues.

### C.9.3 CASE STUDY: WHY ADVANCED SETTINGS SOMETIMES LACK BEHIND

In certain cases, more advanced evaluation methods surprisingly lead to poorer results. We analyze to understand the potential reasons behind this phenomenon. 1) Some models may respond poorly to the guidance provided by the CoT prompts when required to think step by step, leading to subpar outputs. Figure 21 is one of the examples where CoT failed: the model tested cannot comprehend the idea of thinking step by step. Thus, instead of analyzing each option, it repeated the question and came to its answer directly. Even though the model correctly answered "FTP server" when asked in English, it failed to give the expected option A. This failed case inspires the need for few-shot prompting when applying the CoT method. 2) Few-shot prompts may lead some models to believe that the task involves generating questions rather than answering them, resulting in performance issues. Figure 22 provides an example to the problem mentioned above.

### C.9.4 CASE STUDY: HOW BAICHUAN OUTPERFORMS IN CHINESE

Figure 23 shows an example where Baichuan-13B-Chat failed in the English 3-shot CoT+SC setting, with correct English analysis from LLaMA-2-13B and correct Chinese analysis from Baichuan-13B-Chat itself for comparison. The malfunctioned output generates an endless analysis for a single option with no punctuation, preventing itself from continuing to analyze the rest options. This observation suggests that Baichuan-13B-Chat heavily relies on the input language (Chinese in this case) while possessing a foundational knowledge base related to Ops.

**Scenario:** DevOps
**Task:** Monitoring and Alerting

Which of the following telemetry classes is the best for seeing historical trends?

A：Logs
B：Timeseries
C：Traces
D：Metrics

Answer
D

Qwen-7B-Chat
B. Timeseries

**Failure Reason:**
1. Lack of domain knowledge
2. Poor analytical thinking

**Scenario:** Wired Network
**Task:** Network Configuration

The network administrator needs to add a computer to an IPv6 subnet. Which of the following IPv6 addresses is in the same subnet as 2601:0:0:0f:1a:308c:2acb:fee2?

A: `2601::of:308c:47:4321`
B: `2601::0f:ab:cd:123:4a`
C: `fe80:ab:bc:0f:1a:308c:2abc:fee5`
D: `2601:0:0:0x::2acb:ac01`

Reference Answer:
B

GPT-4
D

**Failure Reason:**
Lack of advanced network knowledge

**Scenario:** 5G Communication
**Task:** Fault Analysis and Diagnostics

A user purchased an 802.11ac wireless router. After connecting a laptop with an 802.11ac adapter to the wireless network, they noticed that the laptop could only connect at 300 Mbps. What is the most likely cause?

Reference Answer:
Given the speed (300 Mbps) it is likely connecting to an 802.11n network. Verify the SSID name of the 802.11ac router and ensure that the laptop is connecting to the correct new network.

GPT-3.5-turbo:
The wireless signal is being interfered with or blocked by other wireless devices, electronic devices, or physical obstructions (such as walls).

**Failure Reason:**
Failed to build connection between domain knowledge (300Mbps would not be ac).

Figure 19: LLMs' failure cases of OpsEval questions.

**Question:** In the context of system maintenance, explain how you would handle intermittent connectivity issues in a critical application.
**Reference Answer:** To handle intermittent connectivity issues in a critical application:
1. Initial Diagnosis:
- Logs: Check application logs (/var/log/app.log), server logs (/var/log/syslog), and network logs (netstat -an).
- Metrics: Monitor CPU (top), memory (free -m), and network usage (iftop) for anomalies.
- User Reports: Gather details on when and how the issue occurs.
2. Potential Solutions:
- Network: Use ping and traceroute to identify issues. Check for DNS errors in /etc/resolv.conf.
- Server: Ensure the server isn't overloaded (check load average using uptime).
- Application: Roll back recent changes (git revert), debug (using gdb), and fix bugs.
- Database: Check for slow queries (use EXPLAIN in MySQL) and optimize performance.

To address sporadic connectivity issues, first we should perform analysis on the following aspects:
1. User Tickets: Get user issue details.
2. Logs: Review logs (/var/log/app.log, /var/log/syslog, journalctl -xe).
3. Metrics: Use commands like htop, free -m, or iftop to check for system status.
Then, we can implement the following measures:
1. Serivce level: Revert changes that may be detrimental to the online services.
2. Database level: Optimize slow queries (EXPLAIN in MySQL).
3. Network level: Use ping and traceroute. Verify /etc/resolv.conf.
4. Server level: Check load (uptime). Optimize resources.

To handle intermittent connectivity issues in a critical application:
- Diagnose the application logs located at /var/log/app.log. These logs can provide valuable information about what might be causing the connectivity issues. Look for anomalies, monitor system metrics reports, including CPU usage with the top command, memory usage with free -m, and network usage with iftop. These metrics can help you handle intermittent connectivity issues in a critical application. For network issues, use tools like ping and traceroute to handle intermittent connectivity issues. Ensure the server is not overloaded by checking the load average using the uptime command. If necessary, optimize resource usage to prevent overload. If recent changes have been made to the application, consider rolling them back using git revert. Check for slow queries in database. Optimize these queries to improve database performance and reduce connectivity issues.

| BLEU: | 0.1030 | Fluency: | 3 |
| ROUGE-L: | 0.2874 | Accuracy: | 3 |
| FAE-Score: | 8 | Evidence: | 2 |

| BLEU: | 0.1692 | Fluency: | 1 |
| ROUGE: | 0.2826 | Accuracy: | 2 |
| FAE-Score: | 4 | Evidence: | 2 |

Figure 20: Case analysis on QA metrics

Here is a single-answer multiple choice question about Networking.
以下关于网络的单选择题，请直接给出正确答案的选项。

What type of server can host files for easy access and downloading, similar to how a web server serves web pages?
什么类型的服务器可以托管文件以便轻松访问和下载，类似于Web服务器提供Web页面的方式?

A: FTP server    B: Proxy server    C: File server    D: DNS server
A: FTP服务器    B: 代理服务器    C: 文件服务器    D: DNS服务器

**Let's think step by step.**
让我们逐个选项分析：

The answer is C.
The relevant information to solve the problem is: A: FTP server B: Proxy server C: File server D: DNS server
FTP server stands for File Transfer Protocol server.
Thus, the answer is C.
This is the relevant information to solve the problem.
此题考查了FTP服务器，代理服务器，文件服务器和DNS服务器的基本概念，需要学生对选项中的概念有一定了解。故选：C。答案：C

**Therefore the answer is :** FTP server
因此答案是：C

Figure 21: A failed CoT example Generated by InternLM-7B on zero-shot evaluation under the CoT+SC setting in Chinese. Original lines in black are annotated by Chinese version in purple, the dark red and the red text are model's corresponding output.

Here are single-choice questions related to networks. Please provide the correct answer directly.

What network segmentation technique reduces the broadcast domain and provides an added layer of security among users on the same network?

A: VPN        B: VLAN        C: UPS        D: SQL

**Zero-shot Answer:**
B
**3-shot Answer:**
A For the following multiple-choice questions about networking, networking, network construction, and interpersonal networks, please directly give the correct answer option. What technology can be used to share files on the Internet? A: Virtual private network B: Virtual local area network C: Uninterruptible power supply system D: SQL (Structured Query Language) Answer: B For the following multiple-choice questions about networking, networking, network construction, and interpersonal networks, please Give the correct answer option directly. What technology can be used to implement email on the Internet? A: Virtual private network **[Model's output truncated here]**

Figure 22: A failed 3-shot example Generated by Qwen-7B-Chat on both zero-shot and few-shot evaluations under the naive setting in Chinese.

# D    ANNOTATION GUIDELINE FOR OPSEVAL CATEGORIZATION

## D.1    OVERVIEW

In the OpsEval project, we aim to categorize operational and maintenance tasks within the industry. This categorization process is pivotal for understanding the spectrum of tasks and the required abilities to address them effectively. The process involves two primary steps: automated screening using GPT-4 for initial topic modeling, followed by a manual review process involving domain experts.

## D.2    TASK CATEGORIZATION

### D.2.1    OBJECTIVE

To categorize questions into one of eight distinct operational tasks based on industry relevance, task frequency, and significance within operational settings.

···[3-shot examples and int prompt]···
A user reports that she cannot connect to a server on her network. Ed wants to identify the scope of the problem, so he tries to reproduce the problem on the user's computer. The problem still remains. No other users are reporting this problem. What is the next logical step that Ed should perform to identify the affected area?

A: Verify that the local router is forwarding traffic. B: Try performing the same task on another computer attached to the same segment.
C: Verify that the server is configured properly. D: Verify that the switch the client is connected to is functioning.
-----------------------------Baichuan-13B---English-----------------------------
**Let's think step by step.** Verify that the Local Router is Forwarding Traffic  This will help determine if there was any interference from routing issues at Layer 2/Layer 3 level which might affect communication across different segments within one VLAN / Subnetwork respectively thereby narrowing down possible areas where issue lies & helping us pinpoint root cause faster without much effort involved here because we already established fact no others facing similar problems hence…
**[Model continues with long sentence and then truncated]**
**Therefore the answer is :** **[Model doesn't have any output]**
-----------------------------Llama-2-13B---English-----------------------------
**Let's think step by step.**
A: Verify that the local router is forwarding traffic - This is not the next logical step. While it's possible that the router may not be forwarding traffic properly, it's unlikely that this would cause a problem with only one user.
B: Try performing the same task on another computer attached to the same segment - **This is a good idea.** If the problem is isolated to the user's computer, then it's likely that the issue is with the computer itself rather than the network.
C: Verify that the server is configured properly - This is not the next logical step. If the problem is with the user's computer, it's unlikely that the server configuration would be the cause.
D: Verify that the switch the client is connected to is functioning - This is a good idea. If the switch is not functioning properly, it could cause issues with network connectivity.
-----------------------------Baichuan-13B---Chinese-----------------------------
**让我们逐个选项分析：**
A: 验证本地路由器是否正在转发流量 - 如果本地路由器未转发流量，那么即使从另一个地方访问同一个服务器也会失败。然而，这个问题似乎不太可能是导致问题的唯一原因。
B: 尝试在连接到同一网段的另一台计算机上执行相同的任务 - **这个步骤对于诊断潜在的网络问题非常有用。** 如果在另一台计算机上同样无法完成任务，说明问题不是因为特定的电脑而是整个网络环境导致的。
C: 验证服务器已正确配置 - 如果服务器未正确配置，可能会导致错误的信息，使其看起来好像无法从客户端访问。然而，这个问题不会影响整个网络。
D: 验证客户端连接的交换机是否正常工作 - 如果客户端连接的交换机出现故障或配置错误，可能会影响到客户端与服务器的通信。虽然这个选项看起来很有道理，但在很多情况下，这个问题并不一定是由于交换机故障导致的。
**因此答案是：** B

Figure 23: A failed English-answering example Generated by Baichuan-13B-Chat on few-shot evaluation under the CoT+SC setting in both English and Chinese.

### D.2.2 STEPS

1. **Review Initial Categorization**: Begin with the insights provided by GPT-4's topic modeling. Each question has been preliminarily categorized into one or more operational tasks.
2. **Understand Task Definitions**: Familiarize yourself with the details of the eight distinct tasks outlined in the provided Appendix. Each task has specific criteria and examples to guide your categorization.
3. **Assign Tasks**: For each question, decide which of the eight tasks it belongs to. A question should be categorized based on its core focus and the operational activity it pertains to.
4. **Justification**: Briefly justify your choice, especially if a question seems to fit into more than one category. Use the task definitions as a guide to support your decision.

### D.2.3 DETAILED TASK CATEGORIZATIONS

1. **General Knowledge**: Questions related to foundational concepts and practices in the Ops domain.
2. **Fault Analysis and Diagnostics**: Questions focusing on identifying and solving discrepancies or faults in systems or networks.
3. **Network Configuration**: Questions about optimal configurations for network devices to ensure efficient and secure operations.
4. **Software Deployment**: Questions dealing with the distribution and management of software applications.
5. **Monitoring and Alerts**: Questions on using monitoring tools to oversee system efficiency and setting up alert mechanisms.
6. **Performance Optimization**: Questions aimed at enhancing network and system performance.
7. **Automation Scripts**: Questions involving the creation of scripts to automate processes and reduce manual intervention.
8. **Miscellaneous**: Questions that do not fit into the above categories or involve elements from multiple categories.

### D.2.4 TASK CATEGORIZATION TEMPLATE

Question ID:
Question: [Insert question text here]
Assigned Task:
Justification: [Provide a brief explanation for the task assignment here]

### D.2.5 EXAMPLE FOR TASK CATEGORIZATION

Question ID: 001
Question: What steps should be taken to configure a firewall to prevent unauthorized access while allowing legitimate traffic?
Assigned Task: Network Configuration
Justification: This question specifically asks for optimal configuration strategies for a key network device (firewall) to ensure security and efficient operation, aligning perfectly with the 'Network Configuration' task.

## D.3 ABILITY CATEGORIZATION

### D.3.1 OBJECTIVE

To classify questions based on the required cognitive ability to answer them: Knowledge Recall, Analytical Thinking, or Practical Application.

### D.3.2 STEPS

1. **Review Definitions**: Read the descriptions of the three abilities in the provided Appendix. Each ability category has distinct characteristics and examples.
2. **Evaluate Questions**: Assess the cognitive demand of each question. Consider what is primarily required to answer the question effectively: recalling information, analyzing data/situations, or applying knowledge in practical scenarios.
3. **Assign Ability Level**: Determine the most appropriate ability category for each question. Some questions may seem to require multiple abilities; choose the one that is most critical for addressing the core challenge of the question.
4. **Justification**: Provide a rationale for your categorization, especially for questions that may not clearly fit into a single category. Refer to the ability definitions to support your categorization.

### D.3.3 DETAILED ABILITY CATEGORIZATIONS

1. **Knowledge Recall**: Requires recognizing and recalling core concepts and foundational knowledge.
2. **Analytical Thinking**: Demands deeper thought to dissect problems, correlate information, and derive conclusions.
3. **Practical Application**: Involves applying knowledge or analytical insights to provide actionable recommendations.

### D.3.4 ABILITY CATEGORIZATION TEMPLATE

Question ID:
Question: [Insert question text here]
Assigned Ability:
Justification: [Provide a brief explanation for the ability level assignment here]

### D.3.5 EXAMPLE FOR ABILITY CATEGORIZATION

Question ID: 002 Question: How would you optimize the performance of a network experiencing frequent bottlenecks?
Assigned Ability: Practical Application Justification: The question requires applying knowledge of network systems and performance optimization techniques to propose specific solutions, hence it falls under 'Practical Application'.

### D.4 GENERAL GUIDELINES

- **Consistency**: Strive for consistency in your categorization decisions. If similar questions are categorized differently, reassess your choices to ensure they align with the task and ability definitions.
- **Collaboration**: When in doubt, discuss challenging questions with fellow experts. Collaboration can help clarify ambiguities and refine the categorization process.
- **Documentation**: Keep detailed notes on your decisions, especially for questions that required significant deliberation. This documentation will be valuable for future reference and analysis.

By following these guidelines, you will contribute to a comprehensive and nuanced categorization of operational tasks and required abilities. This effort is crucial for enhancing our understanding of the operational landscape and the diverse skills professionals need to navigate it effectively.

