# Appendix

## A Data sheet

We follow the documentation frameworks provided by Gebru et al. (2018).

### A.1 Motivation

**For what purpose was the dataset created?**

The dataset was created to address the need for an effective benchmark to evaluate the performance of Large Language Models (LLMs) and Ops-specific LLMs (OpsLLMs) in IT operations (Ops) tasks. It aims to inform about the performance of current LLMs on Ops tasks and to aid in optimizing OpsLLMs tailored for the Ops domain. The benchmark, named OpsEval, was designed to tackle challenges such as sensitive data, numerous sub-domains, prompt sensitivity, and appropriate QA metrics in the Ops field.

### A.2 Composition

**What do the instances that comprise the dataset represent (e.g., documents, photos, people, countries)?**

The instances that comprise the dataset represent questions in two formats: multiple-choice questions (MC) and question-answering (QA) questions.

**How many instances are there in total (of each type, if appropriate)?**

7,184 multiple-choice questions and 1,736 QA questions.

**What data does each instance consist of? Is there a label or target associated with each instance?**

Multiple-choice questions consist of a stem, options, and an answer. Question-answering questions consist of a stem and an answer.

**Are relationships between individual instances made explicit (e.g., users' movie ratings, social network links)?**

Questions in OpsEval are relatively independent, but they are clustered based on their classifications (8 tasks and 3 abilities) to group different questions by task type and general capabilities.

**Are there recommended data splits (e.g., training, development/validation, testing)?**

In our evaluation, we use few-shot evaluation. Therefore, we split the data into development (dev) and test sets. Detailed split information can be found in the dataset repository.

**Is any information missing from individual instances? Are there any errors, sources of noise, or redundancies in the dataset? Is the dataset self-contained, or does it link to or otherwise rely on external resources (e.g., websites, tweets, other datasets)? Does the dataset contain data that, if viewed directly, might be offensive, insulting, threatening, or might otherwise cause anxiety?**

No information is missing from individual instances. No errors, sources of noise or redundancies in the dataset. The dataset is self-contained. The dataset does not contain offensive data.

**Does the dataset contain data that might be considered confidential (e.g., data that is protected by legal privilege or by doctor– patient confidentiality, data that includes the content of individuals' non-public communications)?**

The dataset does not contain data that might be considered confidential. We have desensitized the questions in the dataset to ensure that no internal private information from the participating companies is included.

### A.3 Collection Process

**How was the data associated with each instance acquired?**

The data associated with each instance was acquired from four primary sources: company materials, certification exams, Ops textbooks, and automated generation.

Company materials include directly observable data such as Ops tickets and error logs, as well as internal documents and tests for Ops staff training, provided by cooperating companies from various sectors. Certification exam questions were sourced from public study guidebooks for Ops certification exams. Operations textbook data was acquired by searching for relevant books and extracting complete knowledge content and exercises. Automated generation involved using authoritative Ops textbooks and GPT4 to generate diverse questions. Each source is highly esteemed globally and reviewed by our Ops collaborators to ensure data validation and reliability.

**What mechanisms or procedures were used to collect the data (e.g., hardware apparatuses or sensors, manual human curation, software programs, software APIs)?**

The dataset has been assembled from various sources through a combination of manual human curation and automated generation processes. More specifically:

- **Company Materials.** These materials were manually curated by experts from cooperating companies, including production environment materials like Ops tickets and error logs, as well as internal documents and tests for Ops staff training. Experts from 10 companies in sectors such as telecommunications, finance, and Ops service/tool providers contributed to this effort.

- **Certification Exams.** Certification exam questions were sourced from publicly available study guidebooks. These guidebooks, obtained from public book websites, contain knowledge assessments necessary for becoming an Ops staff and are naturally in the form of multiple-choice and question-answering questions. The questions were manually extracted from these guidebooks.

- **Operations Textbooks.** Relevant operations textbooks were identified by constructing a keyword list for the Ops field. These textbooks, which contain comprehensive knowledge content and exercises, were manually reviewed, and relevant questions were extracted for the dataset.

- **Automated Generation.** To enhance the diversity and depth of our test set, we used software programs to extract content from authoritative Ops textbooks and employed GPT-4 through software APIs to generate additional questions. This process involved manual verification by experts to ensure the quality and relevance of the generated questions.

The combination of these mechanisms ensures the reliability and robustness of the data for evaluating LLMs in the Ops domain. Full details, including the sources of the materials, are provided in the documentation accompanying our GitHub repository.

**Who was involved in the data collection process (e.g., students, crowdworkers, contractors) and how were they compensated (e.g., how much were crowdworkers paid)?**

The data collection process involved a team comprising one undergraduate student, three graduate students, and over 20 experts from various companies. All participants, including the crowdworkers, voluntarily contributed to the data collection effort without any financial compensation.

**Over what timeframe was the data collected?**

The data was collected over the timeframe from July 2023 to May 2024, and the collection is still ongoing for the dataset's expansion and maintenance.

### A.4 Preprocessing/cleaning/labeling

**Was any preprocessing/cleaning/labeling of the data done (e.g., discretization or bucketing, tokenization, part-of-speech tagging, SIFT feature extraction, removal of instances, processing of missing values)?**

Please refer to the OpsEval Benchmark section of the paper for details on any preprocessing, cleaning, or labeling of the data.

**Was the "raw" data saved in addition to the preprocessed/cleaned/labeled data (e.g., to support unanticipated future uses)?**

The "raw" data was saved in addition to the preprocessed, cleaned, and labeled data. However, it has not been made publicly available due to the inclusion of some internal company materials.

**Is the software that was used to preprocess/clean/label the data available?**

Yes, the scripts used for preprocessing, cleaning, and labeling the data are provided in the dataset repository.

### A.5 Uses

**Has the dataset been used for any tasks already? If so, please provide a description.**

The dataset has been used to evaluate the capabilities of large language models. For more details, please refer to the paper.

**What (other) tasks could the dataset be used for?**

The dataset could be used for various tasks, including evaluating the performance of Large Language Models (LLMs) and Ops-specific LLMs (OpsLLMs) in IT operations, such as network configuration, error log analysis, and operational knowledge assessments.

**Is there anything about the composition of the dataset or the way it was collected and preprocessed/cleaned/labeled that might impact future uses?**

Given the sensitivity and proprietary nature of some of the source materials, dataset consumers should be cautious about the potential legal and ethical implications of using this data. For example, some data might inadvertently reflect internal company processes or proprietary information.

To mitigate these risks, dataset consumers should:

- Use the data responsibly: Ensure that the data is used only for research and evaluation purposes and not for commercial exploitation.

- Avoid unfair treatment: Be mindful of potential biases in the data that could lead to stereotyping or unfair treatment of individuals or groups.

- Acknowledge data limitations: Recognize and disclose any limitations or biases in the data when publishing results or deploying models trained on this dataset.

**Are there tasks for which the dataset should not be used?**

1. Commercial purposes: Since some data is derived from proprietary company materials, commercial use could result in legal and ethical issues.
2. Sensitive decision-making processes: Avoid using the dataset for making decisions that could significantly impact individuals or groups, such as hiring decisions, without thoroughly evaluating the fairness and bias in the data.

### A.6 Distribution

**How will the dataset be distributed? When will the dataset be distributed? Will the dataset be distributed under a copyright or other intellectual property (IP) license, and/or under applicable terms of use (ToU)?**

The dataset is currently distributed in Huggingface and Github.

**Have any third parties imposed IP-based or other restrictions on the data associated with the instances? Do any export controls or other regulatory restrictions apply to the dataset or to individual instances?**

No third parties have imposed IP-based or other restrictions on the data associated with the instances. No export controls or other regulatory restrictions apply to the dataset or to individual instances.

## A.7 Maintenance

**How can the owner/curator/manager of the dataset be contacted (e.g., email address)?**

The owner/curator/manager of the dataset can be contacted via email. (Our email address will be released once the paper be accepted.)

**Is there an erratum? If so, please provide a link or other access point.**

Currently, there is no erratum. If any errors are found in the future, they will be updated on GitHub. We welcome users to raise issues on GitHub to point out any errors.

**Will the dataset be updated (e.g., to correct labeling errors, add new instances, delete instances)?**

The dataset will be updated at least monthly by the authors. Updates will be announced via GitHub.

**If the dataset relates to people, are there applicable limits on the retention of the data associated with the instances (e.g., were the individuals in question told that their data would be retained for a fixed period of time and then deleted)?**

The dataset does not relate to personal data, so there are no applicable limits on the retention of data associated with individual instances.

**Will older versions of the dataset continue to be supported/hosted/maintained?**

Older versions of the dataset will be maintained in the GitHub history. Relevant updates will be communicated to users via GitHub notifications.

**If others want to extend/augment/build on/contribute to the dataset, is there a mechanism for them to do so?**

We will communicate with contributors and review and process the contributed data following the same curation process outlined in the paper.

# B  Details of OpsEval Benchmark

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

## B.8    Scoring Rubrics of Fluency in FAE-Score

As show in Figure 6, we asked the judge model and experts about the aspects of grammatical correctness, coherence and consistency, calrity of expression, style and tone appropriateness and answer completion of the models' responses.

## B.9    Automated QA generation

During the data collection process, we have experimented automating question-answer generation. We first sampled the QA pairs and manually assessed their accuracy and domain relevance. Later, we used typical manual evaluation examples for few-shot learning, enabling GPT to evaluate QA

Which of the following represents quantifying data moved from one host to another within a specific time frame?
A: Reliability
B: Response time
C: Throughput
D: Jitter
Answer: C
Analysis: Throughput is the measure of data transferred from one host to another in a given amount of time
Task: Performance Optimization
Ability: Knowledge Recall

Which command enables a router to signal clients that they should acquire additional configuration details from a DHCPv6 server?
A: ipv6 nd ra suppress
B: ipv6 dhcp relay destination
C: ipv6 address autoconfig
D: ipv6 nd other-config-flag
Answer: D
Analysis: The **ipv6** nd other-config-flag** command is used to enable a router to inform clients that they need to get additional configuration information from a DHCPv6 server
Task: Automation Scripts
Ability: Analytical Thinking

Question: You receive a call from a user experiencing difficulties connecting to a new VPN. What is the initial step you should take?
A: Find out what has changed.
B: Reboot the workstation.
C. Document the solution.
D: Identify the symptoms and potential causes.
Answer: D
Analysis: Since this is a new connection, you need to start by troubleshooting and identify the symptoms and potential causes
Task: Fault Analysis and Diagnostics
Ability: Practical Application

Figure 3: Three examples of the processed questions

Question: You have a router interface with an IP address of 192.168.192.10/29. What is the broadcast address that the hosts on this LAN will utilize?
问题：路由器上有一个接口，IP地址为192.168.192.10/29。主机在这个局域网上使用的广播地址是什么？

Keypoint: 192.168.192.15
答案要点：192.168.192.15

Detailed Answer: A /29 (255.255.255.248) has a block size of 8 in the fourth octet. This means the subnets are 0, 8, 16, 24, and so on. 10 is in the 8 subnet. The next subnet is 16, so 15 is the broadcast address.
答案解析：/29（255.255.255.248）在第四个八位组有8个块大小。这意味着子网是0，8，16，24等等。10在8的子网中。下一个子网是16，所以15是广播地址。

Task: Network Configuration
任务：网络配置

Ability: Analytical Thinking
能力：推理

Figure 4: An example of the saved subjective questions

A subjective question in OpsEval

Question: You have a router interface with an IP address of 192.168.192.10/29. What is the broadcast address that the hosts on this LAN will utilize?
问题：路由器上有一个接口，IP地址为192.168.192.10/29。主机在这个局域网上使用的广播地址是什么？。

Task: Network Configuration
任务：网络配置

Ability: Analytical Thinking
能力：推理

Prompt

Answer the Reasoning question about Network Configuration.
You have a router interface with an IP address of 192.168.192.10/29. What is the broadcast address that the hosts on this LAN will utilize?
回答关于网络配置的推理问题。
路由器上有一个接口，IP地址为192.168.192.10/29。主机在这个局域网上使用的广播地址是什么？

LLMs

Figure 5: An example of building the prompt of subjective questions.

1. **Grammatical Correctness (0-3 points)**:
• 0: Numerous grammatical errors that hinder comprehension.
• 1: Frequent errors that slightly disrupt the reading flow.
• 2: Minor grammatical errors, but the text remains easily readable.
• 3: Fluent and grammatically correct with no noticeable mistakes.
2. **Coherence and Consistency (0-3 points)**:
• 0: The output is disjointed, lacks logical flow, or contradicts itself.
• 1: Some inconsistencies or a lack of clear logical structure.
• 2: Mostly coherent, though minor clarity issues may be present.
• 3: The response is logically consistent and well-organized.
3. **Clarity of Expression (0-2 points)**:
• 0: The output is vague or ambiguous, making the response unclear.
• 1: Generally clear, though some areas may lack precision or clarity.
• 2: Clear, concise, and directly addresses the question or task.
4. **Style and Tone Appropriateness (0-2 points)**:
• 0: Inappropriate tone for the domain (e.g., overly casual or formal for the task).
• 1: Generally appropriate tone, but occasional mismatches with the task context.
• 2: Consistent tone that is well-suited to the operational context.
5. **Answer Completion (0-2 points)**:
• 0: The response is incomplete or significantly deviates from the expected format.
• 1: Response mostly follows the expected format but misses some details.
• 2: The response fully meets the structural and format requirements of the question.

Figure 6: Scoring Rubrics of Fluency in FAE-Score.

Table 2: Models evaluated in this paper. The "access" column in the table shows whether we have full access to the model weights or can only access them through API.

| Model | Creator | #Parameters | Access | License |
|---|---|---|---|---|
| GPT-4/GPT-3.5-turbo | OpenAI | *undisclosed* | API | Proprietary |
| ERNIE-Bot-4.0 | Baidu | *undisclosed* | API | Proprietary |
| GLM4/GLM3-turbo | Tsinghua Zhipu | *undisclosed* | API | Proprietary |
| Meta-LLaMA-3 | Meta | 8B | Weights | Llama 3 Community |
| LLaMA-2 | Meta | 7/13/70B | Weights | Llama 2 Community |
| Qwen-Chat | Alibaba Cloud | 7/14/72B | Weights | Qianwen LICENSE |
| Qwen1.5-Chat | Alibaba Cloud | 14B | Weights | Qianwen LICENSE |
| InternLM2-Chat | Shanghai AI Laboratory | 7/20B | Weights | Apache-2.0 |
| DevOps-Model-Chat | CodeFuse | 14B | Weights | Apache-2.0 |
| Baichuan2-Chat | Baichuan Intelligence | 13B | Weights | Apache-2.0 |
| ChatGLM3 | Tsinghua Zhipu | 6B | Weights | Apache-2.0 |
| Mistral | Mistral | 7B | Weights | Apache-2.0 |
| Gemma | Google | 2/7B | Weights | Gemma license |
| Claude-3-Opus | Anthropic | *undisclosed* | API | Proprietary |
| Qwen2-Instruct | Alibaba Cloud | 7/72B | Weights | Qianwen LICENSE |

pairs based on our evaluation criteria automatically. Directly generated question-answers tend to be simple judgment or concept questions rather than reasoning questions that better demonstrate the model's capabilities and knowledge density. Our goal is to ensure that while the topics of the questions remain relevant to the seed questions, their specific content is distinct from the original questions. By maintaining the overarching framework in the Ops domain, we can expand the number and types of questions, enabling a more comprehensive evaluation of model capabilities. Additionally, we can incorporate external knowledge during the data generation, continually enhancing our ability to evaluate new content.

## C  Additional details of experiments

### C.1  Detailed Information of LLMs Evaluated

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

Let's think step by step.
让我们逐个选项分析：

A: Routing Information Protocol, version 1 (RIPv1) - This is the correct answer. ···
B: Routing Information Protocol, version 2 (RIPv2) - RIPv2 does include the subnet mask ···
C: Border Gateway Protocol (BGP) - BGP is a routing protocol used for large-scale networks, ···
D: Open Shortest Path First (OSPF) - OSPF supports subnetting and includes the subnet mask ···
A: 路由信息协议，版本1（RIPv1） - 正确。RIPv1不包括子网掩码信息，因此无法支持子网划分。
B: 路由信息协议第二版（RIPv2） - 错误。RIPv2包括子网掩码信息，因此支持子网划分。
C: 边界网关协议（BGP） - 错误。BGP是一种大型互联网路由协议，支持子网划分。
D: 开放最短路径优先（OSPF） - 错误。OSPF是一种内部网关协议（IGP），支持子网划分。

Therefore the answer is : A
因此答案是：A