# OpenReview forum: "OpsEval: A Comprehensive Benchmark Suite for Evaluating Large Language Models’ Capability in IT Operations Domain"
_ICLR.cc/2025/Conference — Submitted to ICLR 2025_

### Official Review · Reviewer_ioEw · 2024-10-31

**Soundness:** 3
**Presentation:** 3
**Contribution:** 3
**Rating:** 5
**Confidence:** 3

**Summary:**

The paper introduces OpsEval, a comprehensive benchmark suite developed to evaluate the capabilities of large language models (LLMs) within the IT operations (Ops) domain. Addressing the need for domain-specific benchmarks, OpsEval encompasses a bilingual multi-task dataset containing 9,070 questions across eight tasks and three cognitive abilities. The benchmark features a novel QA metric, FAE-Score, which surpasses traditional metrics (e.g., BLEU, ROUGE) in alignment with expert evaluations. Results from evaluating over 24 LLMs reveal insights into their performance variability and highlight areas requiring targeted improvements for practical AIOps applications.

**Strengths:**

- **Novelty**: OpsEval is the first comprehensive, bilingual benchmark tailored to IT operations, filling a significant gap in LLM evaluation for this specialized domain.
- **Robust Evaluation Framework**: Introduction of the FAE-Score provides a more reliable assessment method for domain-specific QA, aligning closely with human expert evaluations.
- **Diverse Dataset**: The dataset covers a range of sub-domains and tasks, including multi-choice and QA questions that reflect real-world scenarios, enhancing its practical applicability.
- **Comprehensive Analysis**: The paper presents detailed insights into how different LLMs perform under varied prompting techniques and tasks, providing valuable information for future model improvements and fine-tuning.

**Weaknesses:**

- **Generalizability Concerns**: While the dataset's scope is impressive, its applicability to other IT sub-domains or rapidly changing tech scenarios is not fully addressed.
- **Real-time Adaptability**: The benchmark's utility for models operating in dynamic, real-time environments remains unexplored.
- **Limited Focus on Advanced Reasoning**: Although analytical thinking is a category, the paper does not deeply explore how LLMs handle multi-step or complex reasoning tasks within the Ops context.
- **Dependence on Proprietary Data**: The inclusion of proprietary company data, although necessary, limits the benchmark's accessibility for broader research.

**Questions:**

1. How might OpsEval be adapted for use with real-time or continuously updated data sources?
2. Are there any plans to expand OpsEval to include tasks involving more intricate, multi-step reasoning?
3. What measures are taken to ensure that the benchmark’s results are not biased due to proprietary data contributions?
4. Could the authors provide further detail on the types of error analysis conducted for different models, particularly regarding common failure modes?

---

> ### Author Response · Authors · 2024-11-22
> **Rebuttal**
>
> Dear Reviewer,
>
> Thank you for your insightful feedback and constructive suggestions. Here, we try our best to address your concerns and questions.
> ## W1: Generalizability Concerns
> As the first comprehensive benchmark in Ops domain, we see this work as a foundational starting point. We plan to continuously expand sub-domains and resources through collaboration with the broader Ops community.
>
> Moreover, the classification methodology outlined in Section 3 is designed to be adaptable across various Ops sub-domains. We also believe that the Observations and Practical Lessons summarized at the end of each subsection in Section 4 provide valuable insights that can be applied to scenarios not yet covered by our dataset. These serve as guiding principles for addressing rapidly evolving tech landscapes and new IT sub-domains.
> ## W2: Real-time Adaptability
> Thank you for raising this insightful question. We interpret “models operating in dynamic, real-time environments” as those designed to handle continuously updated data streams, such as real-time monitoring, anomaly detection, or incident resolution in Ops scenarios.
>
> In the Ops domain, there are ML solutions that meet the requirements for dynamic, real-time operations. For LLMs, on the other hand, the current focus should be evaluating whether they possess sufficient contextual knowledge and can fully leverage their strengths in text generation.
>
> In the OpsEval benchmark, the model’s performance on complex QA tasks—such as tool selection, alarm analysis, and task planning—provides an indication of its potential to effectively function as an OpsAgent in the future. This approach lays a foundation for assessing the role LLMs can play in dynamic and real-time operational environments.
>
> ## W3 & Q2: Limited Focus on Advanced Reasoning
>
> We plan to expand OpsEval by designing various task templates, such as ticket analysis, tool selection, and task planning, to further evaluate the model’s reasoning and application capabilities in more intricate, multi-step scenarios. In the revised version, we have added multiple task templates to show how LLMs handle complex reasoning tasks within the Ops context.
>
> ## W4: Dependence on Proprietary Data
>
> In Section 3.1, we introduce the diverse data sources of OpsEval, including company materials, certification exams, and operations textbooks, to ensure the benchmark and dataset are as comprehensive as possible. In our open-source dataset repository, we will provide the distribution of each data source. Questions involving proprietary company data are not entirely withheld from release; instead, they undergo data anonymization and expert review processes to ensure that the benchmark remains accessible for broader research while preserving data privacy and security.
>
> ## Q1.  How might OpsEval be adapted for use with real-time or continuously updated data sources?
> We appreciate your suggestion to cover the adaptability of our framework, we address this problem from two aspects:
>
> Framework Versatility: The OpsEval evaluation framework is designed to have a general API for different models and tasks, for those models following the mainstream of LLMs and questions in standard formats like json and csv, few lines of configuration can achieve the extension; for LLMs with new interface specs, new model class can be written for intergration.
>
> Community Contributions: As part of the AIOps community, OpsEval benefits from and contributes to collective efforts, ensuring ongoing updates and expansions. Since our submission, we've added 10 new models and 1 new collaborator to OpsEval, and we plan to continue adding more models, types of tasks and collarborators in the future.
>
> ## Q2.  Are there any plans to expand OpsEval to include tasks involving more intricate, multi-step reasoning?
>
> Please refer to our reponse to "W3 & Q2: Limited Focus on Advanced Reasoning" sections above.
>
> ## Q3.  What measures are taken to ensure that the benchmark’s results are not biased due to proprietary data contributions?
>
> In Section 6, we conducted a data leakage test, which essentially evaluates whether different models exhibit bias towards the test set distribution. This helps ensure that the benchmark results are not influenced by proprietary data contributions.
>
> ## Q4.  Could the authors provide further detail on the types of error analysis conducted for different models, particularly regarding common failure modes?
>
> Thank you for your suggestive feedback. In the revised version, we will include a detailed discussion of common error modes observed in large models for Ops question answering. Here, we list the most common error modes for demonstation:
> - lack of domain knowledge
> - hallucinations
> - inaccurate reasoning
> - overconfidence in incorrect answers
> We hope this addition can provide a clearer understanding of the challenges and limitations faced by large models in this domain.

---

> ### Author Response · Authors · 2024-11-30
> **Kind Reminder**
>
> Dear Reviewer ioEW,
>
> As the extended discussion period approaches its conclusion, we hope our responses have adequetely addressed the questions and concerns raised in your initial review. We would be happy to continue the discussion if needed. If you find our responses satifsfactory, we kindly request the reviewer to consider adjusting the rating.
>
> Sincerely,
>
> Authors

---

### Official Review · Reviewer_Tvdd · 2024-10-31

**Soundness:** 3
**Presentation:** 4
**Contribution:** 3
**Rating:** 8
**Confidence:** 2

**Summary:**

This paper focuses on IT Operations (Ops) and proposes a comprehensive evaluation benchmark for operations-related large language models (OpsLLMs). The authors initiate a community, involving companies to address privacy challenges. They classify questions under this topic into well-defined categories, providing a dataset of 9,070 examples, with 20% made publicly available for research. Additionally, they introduce a new evaluation metric, FAE-Score, which demonstrates a higher correlation with scores given by human expert compared to traditional metrics like BLEU and ROUGE.

**Strengths:**

- The dataset includes a broad range of categories and question types, accommodating diverse testing setups such as chain of thought (CoT), self-consistency, and few-shot in-context learning.
- The questions and data are collected from the Ops community, reflecting real-world performance issues, making this dataset a valuable resource for researchers in AIOps.
- They propose the FAE-Score, which aligns more closely with human evaluations than BLEU or ROUGE.
- The paper has a clear structure, good writing skills and sufficient figures.

**Weaknesses:**

- The evaluation could benefit more from an analysis of option bias, like testing whether altering the order of choices in questions impacts the results. This could provide insights into any unintended biases within the model and improve the robustness of the evaluation.

**Questions:**

N/A

---

> ### Author Response · Authors · 2024-11-22
>
> Thank you for your thoughtful suggestion regarding the analysis of option bias. We agree that testing for option bias is an important aspect of evaluating model robustness. In Section 6, we conducted a Dataset Leakage Test, which shares a similar objective with option bias analysis, namely assessing whether the model exhibits significant biases toward the dataset distribution. However, our approach, which involves paraphrasing of question content, is more generalizable and can also be applied to QA tasks beyond multiple-choice questions.
>
> That said, we acknowledge the value of explicitly testing for option bias and will incorporate this analysis in future work to further enhance the robustness of our evaluation methodology. Thank you again for highlighting this important perspective.

---

> > ### Comment · Reviewer_Tvdd · 2024-11-26
> >
> > Thanks for the authors' response. I understand that data leakage test provides insight on option bias analysis but there's still model behavior bias brought by prompts. I'll keep my score, but lower my confidence score accordingly.

---

### Official Review · Reviewer_bmT3 · 2024-10-31

**Soundness:** 4
**Presentation:** 3
**Contribution:** 2
**Rating:** 6
**Confidence:** 2

**Summary:**

This paper presents OpsEval, where the authors extract questions related to IT operations from multiple sources, and after preprocessing, automated labeling, and expert review, they construct a dataset containing 9070 entries. During LLMs evaluation, the authors propose the FAE-Score to address the limitations of the rouge and bleu scores, demonstrating a high correlation with human evaluation. Additionally, the paper conducts extensive experiments to test the performance of various base LLMs across different prompt types, task categories, and languages, and analyzes the underlying reasons for the results.

**Strengths:**

- The paper introduces a novel dataset, filling the gap in benchmarks for the IT operations domain.
- This is a highly comprehensive piece of work. Each step, from dataset construction to LLMs evaluation, is thoroughly executed, including data labeling and experimental comparisons.
- The FAE-Score shows better results compared to existing evaluation metrics, with the potential to become a widely adopted metric.

**Weaknesses:**

- There are some minor issues with the paper's writing. For example, the same figure appears twice on page 14.
- For one of the main innovations, the FAE score, the paper only provides a textual description, without a formal definition or concrete examples (e.g., keyword extraction and similarity search).

**Questions:**

1. Is there any plan to further explore and use Automated QA generation? This is a direction that holds significant potential for deeper exploration.
2. Regarding the fluency score, Figure 11 presents a total score of 12 points, but the expert rating is between 0-3. Is there a conversion process involved here?
3. How to prove that taking fluency and evidence into account is reasonable? For multiple-choice questions, answering correctly is the most important factor, but is it possible that the FAE-Score might give a higher score to a response that is very logical but incorrect?

---

> ### Author Response · Authors · 2024-11-22
> **Rebuttal**
>
> Dear Reviewer,
>
> Thank you for your insightful feedback and constructive suggestions. Here, we try our best to address your concerns (W) and questions (Q).
>
> ## W1: Minor writing issues
> We have removed the duplicate figure in the revised version. We sincerely appreciate the feedback and will continue to refine our paper's writing.
>
> ## W2: FAE-Score Details
> Thank you for your valuable suggestion. In the revised version, we have updated Figure 3 to include a concrete example as illustrations of FAE-Score. Additionally, we have supplemented Section 3 with formal definitions of the metrics and detailed explanations of key components, including Keyword Extraction and Similarity Search. We hope these additions will provide a clearer understanding of our work.
>
> ## Q1: Discussion on automated QA generation
> We fully agree with the reviewer's insight on automated QA generation. We are designing task-specific templates tailored to the characteristics of Ops documents, enabling LLMs to generate QA tasks based on the content of these documents. These tasks include, but are not limited to, ticket summarization, code generation, and reasoning analysis. We first construct the QA generation prompt
> In the revised version, we have incorporated this discussion into the revised version of the paper under the Discussion section, and have added some example task templates in the Appendix to illustrate how we attempt to generate Ops-related QAs.
>
> ## Q2: Conversion of fluency score
> When calculating the total FAE score for comparison with the expert score, we scaled the Fluency score to ensure equal weighting among the three metrics. However, when evaluating the correlation between the Fluency metric and the expert score alone, multiplying or dividing by a constant does not affect the correlation coefficient.
>
> ## Q3: Rationale of fluency and evidence metrics
> The FAE-Score is specifically designed for evaluating open-ended QA tasks rather than multiple-choice questions. As described in Section 3.3, we use exact match as the evaluation method for multiple-choice questions. While multiple-choice questions are easier to design and evaluate, we believe that for complex Ops tasks, developing open-ended QA questions and appropriate evaluation metrics is the key direction for this benchmark’s growth and its ability to address real-world challenges.

---

> ### Comment · Reviewer_bmT3 · 2024-11-25
>
> Thanks for the response and the revision of the paper. Overall, I think most of my issues have been solved and I have updated my score. But to be honest, I am not an expert in this field, so there may be some mistakes in my judgements, so I will lower my confidence score accordingly.

---

### Official Review · Reviewer_pP8W · 2024-11-04

**Soundness:** 2
**Presentation:** 2
**Contribution:** 2
**Rating:** 3
**Confidence:** 4

**Summary:**

This paper presents the OpsEval benchmark for assessing LLMs in IT operations (Ops). Moreover, a new evaluation method for QA in Ops is also introduced. Over 24 LLMs are evaluated in the proposed benchmark.

**Strengths:**

1. Introduced a new benchmark containing 9070 questions to evaluate LLMs in IT operations. The proposed benchmark is also diverse as it covers 8 tasks and three abilities across various IT sub-domains.

2. The proposed FAE-Score evaluation metrics that focused on fluency, accuracy, and evidence outperform other existing evaluation metrics in terms of alignment with humans.

3. Experiments were conducted over 24 LLMs to evaluate their capabilities and limitations in this benchmark with various prompting techniques, quantization, etc.

4. The authors also release a small subset of the OpsEval benchmark.

**Weaknesses:**

1. The open-source version is quite limited in size, which makes the contribution quite narrow in scope.
2. The approach to select the 20% data is not discussed in the paper.
3. The contribution in the paper is quite incremental. There are existing relevant benchmarks like NetOps and OWL. The authors claimed that their contribution in this paper in comparison to these existing benchmarks is the continuously updated leaderboard. This cannot be considered as a major contribution.
4. The data from real-world companies are also quite small in comparison to other domains like Wired network or 5g Communication. This makes the usefulness of the proposed benchmark for real-world practical scenarios quite limited.
5. The technical contribution in the FAE metric is quite limited. It just seems a utilization/combination of various existing techniques. Also, the motivation behind using various techniques is not clearly demonstrated. For instance, why QWEN was selected to evaluate fluency in FAE?
6. The authors criticized ROUGE and BLEU in various sections of the paper, then why did they adopt a ROUGE based method in the FAE metric for evidence evaluation?
7. Moreover, there are also contextualized metrics like AlignScore or BERTScore. Therefore, why these metrics were not used as an alternative to ROUGE?

**Questions:**

See above.

---

> ### Author Response · Authors · 2024-11-22
> **Rebuttal**
>
> Dear Reviewer,
>
> Thank you for your insightful feedback and constructive suggestions. Here, we address your concerns and questions.
>
> ## W1: Limited open-source dataset
> We appreciate the reviewer’s feedback regarding the scope of the open-source version of the dataset. We would like to clarify that the dataset is continuously being expanded, and we are committed to releasing more data to further benefit the community as the scale of our dataset continues to grow.
> As described in Section 3.5, the decision of opensourcing 20% data was made to balance openness with ensuring the integrity of the benchmark. The released subset has been carefully curated to provide researchers with valuable insights into the dataset’s structure, question types, and coverage while enabling local evaluations to accelerate model development and iteration.
>
> ## W2: Approach to select the 20% data
> The criterion to selecting the 20% data is based on both the balance of each source and license agreement. We first performed uniform random sampling on different data sources. For the company’s proprietary data, we reached an agreement with the  company to ensure that the sampled data does not contain any private information. We thank the reviewer for pointing out the absense of this discussion, and we will elaborate our open-sourced data selection policy in the revised version (Section 3.5).
>
> ## W3: Related work comparison
> We would like to clarify that the contribution of OpsEval in comparison to NetOps and OWL is not limited to continuously updating leaderboard, as is discussed in the related work section and shown in Table 1, NetOps also lacks a boarder scope of the Operations topics while focusing on wired network operations, and OWL failed to provide a well-designed evaluation methodology for IT operations QA tasks.
>
> ## W4: Limited real-world companies data
> We agree with the reviewer’s perspective on the importance of incorporating more practical data from real-world companies. Expanding the dataset in this direction is a key focus of our ongoing efforts, and we are actively collaborating with our partners to include more question data derived from real-world Ops scenarios.
>
> We would like to point out that the constraints of privacy and security considerations are inherent challenges in Ops field, and do not compromise the representativeness and practical value of the dataset. Additionally, it is important to note that the scale of evaluation dataset should not be directly equated with the scale of training data. Our work focuses heavily on benchmarking, aiming to establish a robust evaluation framework rather than solely providing a large-scale dataset.
>
> ## W5: Details of FAE-score
> The FAE-score is not just a combination of existing techniques but a customized framework designed specifically for Ops-related QA tasks. Its innovation lies in addressing the unique challenges of the Ops domain by comprehensively evaluating the fluency, accuracy, and evidential support of model responses to meet the needs of operations personnel in real-world scenarios.
>
> We appreciate the reviewer’s inquiry regarding the motivation behind our choice of techniques. In the revised version, we have provided further details about the FAE-score in the Section 3.4. Specifically, for Fluency evaluation, we selected Qwen as the base model because of its strong performance in general language generation [1] and its consistent multilingual capabilities without significant degradation. This characteristic is critical for our benchmark, which must account for both Chinese and English scenarios.
>
> ## W6 & W7: ROUGE and other possible metrics
> In the paper, we mentioned that BLEU and ROUGE are not suitable for identifying keywords closely related to Ops tasks, which can lead to scores being influenced by many irrelevant factors when evaluating model outputs. However, for the evaluation of Evidence, the primary focus is on whether the model’s response includes content from the relevant documents to support its conclusions. In this case, ROUGE, by its nature of calculating lexical recall, can effectively reflect the model’s capability in this aspect. Therefore, we chose ROUGE to compute the Evidence metric.
>
> Metrics like AlignScore and BERTScore address the issue of **factual consistency**. In our approach, ROUGE is used to evaluate the **recall rate of relevant documents**, which serves a different purpose. For accuracy, AlignScore and BERTScore could indeed be effective metrics. However, unlike factual consistency in general domains, Ops tasks in QA have more explicit keywords that serve as criteria for determining accuracy (refer to Figure 12). Therefore, we opted for a keyword extraction approach rather than relying on existing general-domain factual consistency metrics.

---

> > ### Comment · Reviewer_pP8W · 2024-11-25
> >
> > Thanks to the authors for their response. I encourage the authors to revise the paper accordingly with more details.

---

> > > ### Author Response · Authors · 2024-11-30
> > > **Kind Reminder**
> > >
> > > Dear Reviewer pP8W,
> > >
> > > As the extended discussion period approaches its conclusion, we hope our responses have adequetely addressed all the questions and concerns raised in your initial review. In the common response *Summary of Key Revisions*, we summarize the key revisions in our paper and how they are intended to address your concerns. We would be happy to continue the discussion if needed. If you find our responses satifsfactory, we kindly request the reviewer to consider adjusting the rating.
> > >
> > > Sincerely,
> > >
> > > Authors

---

### Author Response · Authors · 2024-11-30
**Summary of Key Revisions**

**Dear Program Chairs, Area Chairs and Reviewers,**

We appreciate all the feedbacks from reviewers and have revised our paper accordingly, here we list the major and minor changes in the revised version and explain how they address reviewers' concerns and questions.

1.	**FAE-Score Details (pP8W, bmT3):** In `Section 3.4`, we made following changes to address reviewers' questions about the details of FAE-Score.
	- `Figure 3` is updated to include concrete examples illustrating the FAE-Score.
	- `Section 3.4` now includes formal definitions of the metrics and detailed explanations of key components, such as Keyword Extraction and Similarity Search. For Fluency evaluation, we explained the rationale for choosing Qwen as the base model, citing its strong multilingual performance and suitability for Chinese and English scenarios.
2.	**Data Selection Policy (pP8W)**: In `Section 3.5`, we clarified the criteria for selecting the 20% open-sourced data, ensuring a balance across different sources and compliance with license agreements. For proprietary company data, we confirmed that the sampled data does not include private information.
3.	**Automated QA Generation and Task Templates (bmT3, ioEw):** In `Section 6 (Discussion)`, we discussed the automated QA generation as well as limitation and future direction of our work.
	- In `Appendix`, we have added the prompt for automated QA generation and task-specific templates tailored to Ops documents, to demonstrate how LLMs handle intricate, multi-step scenarios within the Ops context.
5.	**Error Analysis (ioEw)**: In `Section 6`, in response to the reviewer’s suggestion, we included a discussion of common error modes observed in large models for Ops question answering. These include lack of domain knowledge, hallucinations, inaccurate reasoning, and overconfidence in incorrect answers.
6.	**Writing Improvements (bmT3)**: Minor writing issues, such as the removal of duplicate figures, have been addressed. We have also refined the overall writing quality for clarity and precision.

---

### Author Response · Authors · 2024-11-30
**Contributions and Strengths Summary**

We sincerely thank all the reviewers for their constructive feedback and recognition of our work. Here, we summarize key contributions of our work, referencing both the reviewers’ comments and the original statements in our paper.

1. **Introduction of OpsEval, a Novel Benchmark for IT Operations** (all reviewers)：OpsEval is the first comprehensive, bilingual benchmark tailored for IT operations (Ops), addressing a significant gap in evaluating LLMs for domain-specific tasks.
2. **Diverse and Representative Dataset** (all reviewers)：The dataset includes 9,070 questions across eight tasks and three cognitive abilities, derived from diverse sources such as company materials, certification exams, and textbooks, ensuring real-world applicability.
3. **Introduction of the FAE-Score Metric** (all reviewers): We propose FAE-Score, a new evaluation metric focusing on fluency, accuracy, and evidence, which aligns closely with human expert evaluations and outperforms traditional metrics like BLEU and ROUGE.
4. **Comprehensive Evaluation of LLMs and Practical Insights** (pP8W, bmT3, ioEw): The benchmark evaluates 24 LLMs using various prompting techniques, task categories, and quantization strategies. Insights from these evaluations provide guidance for model improvement, fine-tuning, and applicability to sub-domains.
5. **Practical Applicability and Community Engagement** (Tvdd, ioEw): OpsEval supports the AIOps community by addressing privacy challenges and making 20% of the dataset publicly available for research. It provides a robust framework for evaluating domain-specific capabilities, facilitating collaboration and innovation in AIOps research.

---

### Meta-Review · Area_Chair_GXfB · 2024-12-22

**Metareview:**

(a) Scientific Claims and Findings
The paper proposes OpsEval, a benchmark suite tailored for Ops-related LLM evaluation. It features a diverse dataset, a novel FAE-Score metric for QA evaluation, and results from testing 24 LLMs. Key claims include superior alignment of FAE-Score with human evaluations and insights into LLM applicability for domain-specific tasks.

(b) Strengths
- Novel Benchmark: OpsEval addresses the gap in evaluating LLMs for Ops tasks, offering a domain-specific, bilingual dataset.
- FAE-Score: Demonstrates strong correlation with human evaluations, outperforming traditional metrics.
- Comprehensive Evaluation: Tests 24 LLMs using varied setups, providing actionable insights.

(c) Weaknesses
- Limited Open-Source Data: Only 20% of the dataset is available, limiting broader accessibility.
- Incremental Innovation: Some contributions, such as the FAE-Score, build on existing techniques with limited novelty.
- Comparative Analysis Gaps: Insufficient discussion on alternatives like BERTScore and AlignScore.
- Generalizability Concerns: Uncertain applicability to dynamic, real-time Ops scenarios or other IT sub-domains.
- Limited Advanced Reasoning: Does not deeply explore complex reasoning capabilities of LLMs.

(d) Decision: reject

While OpsEval makes valuable contributions as a benchmark for a niche domain, its limited dataset accessibility and incremental innovations reduce its impact. The paper is a solid step forward but falls short of groundbreaking contributions in evaluation methodologies or dataset design.

**Additional Comments On Reviewer Discussion:**

Reviewer Concerns:

- Dataset Accessibility: Reviewers pP8W and ioEw flagged the limited open-source data and questioned its selection criteria.
- Metric Justification: Questions about FAE-Score's innovation and the choice of techniques like ROUGE for evidence evaluation were raised.
- Real-world Applicability: Reviewers ioEw and Tvdd pointed out the lack of real-time adaptability and advanced reasoning tasks.
- Writing Quality: Reviewer bmT3 noted duplicate figures and unclear metric definitions.

Author Responses:

- Clarified data selection policies and committed to expanding the dataset.
- Provided additional FAE-Score details, examples, and rationale for technique choices.
- Addressed feedback with minor revisions, including adding task templates and error analysis.

Review Score Changes:

- bmT3 upgraded their score slightly but lowered confidence.
- Other reviewers acknowledged the responses but did not revise scores significantly.

---

### Decision · Program_Chairs · 2025-01-22

Reject